# THEORETICAL GUARANTEES FOR CAUSAL DISCOVERY ON LARGE RANDOM GRAPHS

**Mathieu Chevalley**[1,2†]    **Arash Mehrjou**[1,3]    **Patrick Schwab**[1]
[1]GSK.ai    [2]ETH Zürich    [3] MPI for Intelligent Systems

## ABSTRACT

We investigate theoretical guarantees for the *false-negative rate* (FNR)—the fraction of true causal edges whose orientation is not recovered, under single-variable random interventions and an $\epsilon$-interventional faithfulness assumption that accommodates latent confounding. For sparse Erdős–Rényi directed acyclic graphs, where the edge probability scales as $p_e = \Theta(1/d)$, we show that the FNR concentrates around its mean at rate $O\left(\frac{\log d}{\sqrt{d}}\right)$, implying that large deviations above the expected error become exponentially unlikely as dimensionality increases. This concentration ensures that derived upper bounds hold with high probability in large-scale settings. Extending the analysis to generalized Barabási–Albert graphs reveals an even stronger phenomenon: when the degree exponent satisfies $\gamma > 3$, the deviation width scales as $O\left(d^{\beta-1/2}\right)$ with $\beta = 1/(\gamma - 1) < 1/2$, and hence vanishes in the limit. This demonstrates that heterogeneous, heavy-tailed degree structures commonly observed in empirical networks can intrinsically regularize causal discovery by reducing variability in orientation error. These finite-dimension results provide the first dimension-adaptive, faithfulness-robust guarantees for causal structure recovery, and challenge the intuition that high dimensionality and network heterogeneity necessarily hinder accurate discovery. Our simulation results corroborate these theoretical predictions, showing that the FNR indeed concentrates and often vanishes in practice as dimensionality grows.

## 1 INTRODUCTION

Causal discovery aims to recover directed acyclic graph (DAG) structures that encode cause–effect relationships among variables. In systems biology, for instance, reconstructing gene regulatory networks from interventional single-cell data provides insight into the mechanisms driving cellular processes (Dixit et al., 2016; Meinshausen et al., 2016; Chevalley et al., 2025a;b). Similarly, in neuroscience, mapping the causal structure of brain activity supports the understanding of functional connectivity and its alterations in disease (Smith et al., 2011; Park & Friston, 2013). A hallmark of such applications is their high dimensionality: typical datasets involve hundreds to thousands of variables. This setting motivates a need for theoretical guarantees that reflect practical constraints and typical graph structures encountered in real-world systems.

A substantial body of work has developed worst-case bounds for causal discovery under interventions, often focusing on the number of experiments required to recover the full DAG (Eberhardt et al., 2005; Shanmugam et al., 2015; Kocaoglu et al., 2017a). However, such analyses assume idealized conditions—e.g., full faithfulness, causal sufficiency, or adversarial intervention designs—that limit their applicability in large-scale settings. Moreover, they do not capture the variability of discovery accuracy across random instances, nor do they account for structural features like degree heterogeneity or sparsity commonly observed in empirical networks (Barabási & Albert, 1999).

In this work, we analyze the performance of interventional causal discovery under more realistic assumptions and with a focus on the *false-negative rate* (FNR), defined as the proportion of true edges whose orientation is not recovered. We adopt an $\epsilon$-interventional faithfulness assumption (Chevalley et al., 2025c) that allows for latent confounding and does not require the full faithfulness property to hold. Interventions are assumed to be selected at random, mimicking experimental designs where

---

† Correspondence to `m.chevalley97@gmail.com`

only limited control over perturbations is feasible. Our focus is on deriving finite-dimension deviation bounds that quantify how tightly the FNR concentrates around its expectation in large graphs.

A wide range of real-world networks exhibit structural features that are well captured, at least approximately, by classical random-graph ensembles such as Erdős–Rényi (Erdős et al., 1960) and Barabási–Albert (Barabási & Albert, 1999) models. Gene regulatory, protein–interaction, and metabolic networks typically display strong sparsity, heterogeneous node degrees, and hub-dominated organization, with numerous studies reporting approximately heavy-tailed or truncated power-law degree distributions (Albert, 2005). Similar patterns arise in neural and brain connectivity networks, where structural and functional connectomes consistently show hub regions and fat-tailed degree behavior (Achard et al., 2006; Eguiluz et al., 2005). Although real networks are not generated by these models exactly, ER and BA distributions capture the dominant combinatorial regularities, such as sparsity levels, neighborhood sizes, and degree heterogeneity. They therefore provide an analytically tractable proxy for understanding how these properties influence the stability of causal discovery at scale.

Our analysis yields several key insights. First, we show that in dense Erdős–Rényi DAGs—where the edge probability $p_e$ is constant—the FNR vanishes in expectation at rate $\Theta(1/d)$ and concentrates with typical width $O(d^{-1/2})$. Thus, in dense random graphs the orientation error not only goes to zero on average, but also becomes sharply localized, providing strong guarantees for reliable causal discovery at large scale. Second, in sparse Erdős–Rényi DAGs—where the number of edges scales linearly with the number of nodes—the FNR remains bounded ($O(1)$) in expectation for a non-vacuous constant, and concentrates at rate $O(\frac{\log d}{\sqrt{d}})$. This implies that even under relaxed assumptions and partial interventions, the orientation error becomes increasingly stable in high dimensions. Third, we extend the results to generalized Barabási–Albert models, which generate scale-free DAGs reflecting the degree distributions of empirical networks. In this case, when the degree exponent satisfies $\gamma > 3$, we prove that the deviation from the expected FNR *vanishes* asymptotically, confirming that scale-free topologies regularize causal inference by suppressing error variability. These findings are surprising: contrary to common beliefs that high dimensionality and structural heterogeneity hinder reliable discovery, we show that they can in fact improve its statistical robustness.

Beyond their theoretical interest, these results are useful to both methodologists and practitioners. For methodologists, the deviation bounds provide a principled baseline for understanding which structural regimes make causal orientation intrinsically easier or harder. This baseline is valuable in two complementary ways. First, it sets an upper bound on the stability guarantees one should expect under more optimistic assumptions, richer intervention designs, or additional structural priors: any method leveraging such information should exhibit deviation behavior that is at least as good as, and typically better than, the guarantees we derive under minimal conditions. Second, it provides a sound reference point for empirical evaluation: synthetic benchmarks routinely use ER, BA, or related random-graph families, and our deviation bounds clarify what variability in FNR one should expect in these settings, as well as how such variability should scale with dimension. For practitioners, the results offer guidance on randomized intervention designs: when performance variability can be assessed on moderate-sized systems, our bounds imply that (in terms of FNR) the variability will typically remain comparable or improve at larger scale due to stronger concentration. This provides reassurance about the behavior of the Intersort score under random interventions and offers a principled way to reason about how many such interventions are required to achieve a desired reliability level.

While prior work has focused on identifiability or asymptotic consistency, ours is the first to establish finite-dimension deviation guarantees—showing that causal orientation errors are not only small in expectation but also sharply concentrated in large-scale, structured settings. To our knowledge, these are the first deviation inequalities for topological errors in interventional causal discovery that (i) hold under minimal $\epsilon$-interventional faithfulness, (ii) capture the distributional properties of widely used random graph ensembles reflecting common structural regimes, (iii) yield dimension-adaptive, variance-sensitive guarantees for false-negative error. By linking graph structure and intervention probability to finite-sample reliability, our results provide both new theoretical insight and practical guidelines for designing interventional studies, and are further corroborated by extensive simulations.

Table 1: **Summary of theoretical results** Asymptotic orders for the expected topological error and the *concentration rate* (typical deviation scale around the mean) in the three random-graph regimes considered. For the generalized Barabási–Albert (BA) model, $\beta = \frac{1}{\gamma-1}$ with $\gamma = 2 + \kappa/m$, $\kappa > 0$. $f$ measures the number of misorientations in the predicted causal order, whereas $g$ is normalized by the number of true edges (*false-negative rate* (FNR)). The probability for a variable to be intervened on is denoted $p_{int}$.

| Model | $\mathbb{E}[f]$ | $\mathbb{E}[g]$ | **Dev.** $f$ | **Dev.** $g$ |
|---|---|---|---|---|
| ER dense ($p_e = \Theta(1)$) | $O\left(\frac{(1-p_{int})^2}{p_{int}}d\right)$ | $O\left(\frac{2(1-p_{int})^2}{p_{int}d}\right)$ | $O(d^2)$ | $O(d^{-1/2})$ |
| ER sparse ($p_e = c/d$) | $O\left(\frac{(1-p_{int})^2}{p_{int}}d\right)$ | $O\left(\frac{2(1-p_{int})^2}{c\,p_{int}}\right)$ | $O(\sqrt{d}\log d)$ | $O(d^{-1/2}\log d)$ |
| BA ($\beta \in (0,1)$) | $(1-p_{int})^2 md$ | $(1-p_{int})^2$ | $O(d^{\beta+1/2})$ | $O(d^{\beta-1/2})$ |

## 2 RELATED WORK

Classical results in interventional causal discovery analyze the *worst-case* number of interventions needed for full graph recovery. For multi-variable interventions, $\mathcal{O}(\log d)$ sets suffice (Eberhardt et al., 2005), while for atomic interventions $d - 1$ suffice (Eberhardt et al., 2006). Subsequent refinements related this complexity to graph structure, e.g. $\mathcal{O}(\log \omega(\mathcal{G}))$ in terms of clique size (Hauser & Bühlmann, 2014). These analyses adopt an adversarial view: the intervention strategy must succeed against any possible graph. (See Appendix A for a more detailed survey of refinements based on randomization (Eberhardt, 2010; Hu et al., 2014) and minimax lower bounds (Shanmugam et al., 2015; Kocaoglu et al., 2017a; Squires et al., 2020a; Porwal et al., 2022).)

A parallel line of work studies *intervention design under constraints*, e.g. budgeted or adaptive designs, where the goal is to maximize utility given a limited number of experiments (He & Geng, 2008; Hyttinen et al., 2013; Ghassami et al., 2018; Hauser & Bühlmann, 2014; Sussex et al., 2021; Agrawal et al., 2019). Here the focus is typically on approximate greedy algorithms rather than fundamental limits.

Beyond worst-case identifiability, another thread studies *interventional Markov equivalence classes* ($\mathcal{I}$-MECs): the partially identified structures remaining after a given set of interventions. Seminal work by Hauser & Bühlmann (2012) introduced $\mathcal{I}$-MECs and greedy algorithms for learning with interventional data. Subsequent work extended these ideas to more general settings (Yang et al., 2018; Jaber et al., 2020; Kocaoglu et al., 2019; Squires et al., 2020b) and proposed efficient learning methods (Brouillard et al., 2020; Ke et al., 2019; 2023; Mooij et al., 2020; Nazaret et al., 2023; Wang et al., 2017; Faria et al., 2022; Lorch et al., 2022). (See Appendix A for a fuller survey.)

However, existing work leaves open a key question: what is the *expected* size of an interventional Markov equivalence class (I-MEC)? This expectation depends not only on the intervention policy but also on the underlying graph distribution. Most prior results emphasize worst-case scenarios—e.g., bounded-degree graphs—which yield robust guarantees but can be overly conservative. By contrast, an average-case analysis under parameterized graph models can provide more realistic insight. For example, many empirical networks in systems biology exhibit heterogeneous, hub-dominated degree patterns, which tend to shrink I-MECs far more than worst-case analyses suggest. Incorporating such stylized but analytically tractable graph ensembles into the theory therefore refines our understanding of intervention utility and provides more practically relevant guidance for experimental design.

Two works that have aimed at analyzing the expected size of $\mathcal{I}$-MECs are Katz et al. (2019) and Chevalley et al. (2025c), both considering single-variable interventions. Under an Erdős–Rényi graph distribution (Erdős et al., 1960), Katz et al. (2019) derive upper bounds for various metrics—such as the number of unoriented edges and the logarithm of the $\mathcal{I}$-MEC size—under the assumptions of causal sufficiency and faithfulness, and assuming an optimal intervention selection policy (i.e., selecting interventions that minimize the number of unoriented edges). They show that, as the number of variables $d$ tends to infinity, these metrics converge to well-defined limits, and they provide finite-$d$ upper-bounds that can be estimated using Monte Carlo simulations.

In contrast, Chevalley et al. (2025c) also analyze Erdős–Rényi graphs but relax some of these assumptions by replacing causal sufficiency and strict faithfulness with an $\epsilon$-interventional faithfulness assumption under a task agnostic selection policy, where interventions are selected

at random. They define a score on *causal orders* that leverages the statistical distances from the interventional data.The causal orders that maximize this score form an equivalence class for which they derive closed-form upper bounds on the expected number of *misoriented edges* (i.e., true edges that do not follow the causal order).

In this work, we extend these theoretical results by analyzing another metric: the false negative rate (FNR), defined as the number of misoriented edges divided by the total number of edges. More importantly, we derive deviation bounds from the mean of those metrics, offering a more detailed perspective on the performance variability of individual instances compared to the expected performance. We also extend all those results to a generalized Barabási–Albert graph distribution (Barabási & Albert, 1999), whose scale-free behavior more closely mimics real-world networks. To the best of our knowledge, our work provides the first deviation bounds for causal discovery errors, as opposed to worst-case or asymptotic consistency results. These bounds hold under $\epsilon$-interventional faithfulness and scale favorably with graph size and topological heterogeneity.

## 3 DEFINITIONS AND NOTATIONS

In this section, we introduce the main notions and assumptions that underlie our analysis, following the formulation in (Chevalley et al., 2025c). In our setting, the causal graph is represented as a directed acyclic graph (DAG) $\mathcal{G} = (V, E)$ over a set of $d$ random variables $\mathbf{X} = (X_1, \ldots, X_d)$ with $V = \{1, \ldots, d\}$. The matrix $\mathbf{A}^{\mathcal{G}}$ denotes the corresponding adjacency matrix, with $\mathbf{A}^{\mathcal{G}}_{ij} = 1$ if $(i, j) \in E$ and 0 otherwise. For any node $j$, the parent set is defined as $\mathrm{Pa}(j) := \{i \in V : (i, j) \in E\}$, and the sets of ancestors and descendants of a node $i$ are denoted by $\mathrm{Anc}_{\mathcal{G}}(i)$ and $\mathrm{Desc}_{\mathcal{G}}(i)$ respectively.

In the context of structural causal models (SCMs) $\mathcal{C} = (\mathbf{S}, P_N)$, each variable $X_j$ is generated by a structural equation $S_j \in \mathbf{S}$ $S_j : X_j = f_j\Big(\mathbf{X}_{\mathrm{Pa}(j)}, N_j\Big)$, where $N_j$ is an exogenous noise variable. We consider *interventions* that replace the original structural equation of a variable $X_k$ by $X_k = \tilde{N}_k$. Given a set of intervention targets $\mathcal{I} \subseteq V$, let $\mathbf{I} = (I_1, \ldots, I_d) \in \{0, 1\}^d$ be the corresponding intervention indicator vector, such that $I_j = 1$ if and only if $j \in \mathcal{I}$. We denote by $P_X^{\mathcal{C},(\emptyset)}$ the observational distribution and by $\mathcal{P}_{int} = \{P_X^{\mathcal{C},do(X_k := \tilde{N}_k)}, k \in \mathcal{I}\}$ the set of interventional distributions.

A causal order is any permutation $\pi : V \to V$ such that for every edge $(i, j) \in E$ we have $\pi(i) < \pi(j)$. To measure the divergence of an ordering $\pi$ from the true causal ordering of $\mathcal{G}$, we use the *top order divergence* (Rolland et al., 2022) defined by

$$D_{\mathrm{top}}(\mathcal{G}, \pi) = \sum_{\pi(i) > \pi(j)} \mathbf{A}^{\mathcal{G}}_{ij}. \tag{1}$$

This score measure the number of misoriented edges given a permutation. Given the observational and interventional distributions, Chevalley et al. (2025c) propose a score function to evaluate the quality of a candidate causal order. For a statistical distance $D : \mathcal{P}(\mathcal{M}) \times \mathcal{P}(\mathcal{M}) \to [0, \infty)$ and parameters $\epsilon > 0$ and $c > \epsilon$, the score function is defined as

$$S(\pi, \epsilon, D, \mathcal{I}, P_X^{\mathcal{C},(\emptyset)}, \mathcal{P}_{int}, c) = \sum_{\substack{i \in \mathcal{I}, j \in V \\ \pi(i) < \pi(j)}} \Big(D_{ij} - \epsilon\Big) + c \cdot d \cdot \mathbf{1}\Big\{D_{ij} > \epsilon\Big\}, \tag{2}$$

where $D_{ij} = D\big(P_{X_j}^{\mathcal{C},(\emptyset)}, P_{X_j}^{\mathcal{C},do(X_i := \tilde{N}_i)}\big)$. This score quantifies how well the ordering $\pi$ aligns with interventional effects, and it is used to define the optimal causal order $\pi_{\mathrm{opt}}$ as the maximizer of the score. The main practical limitations of derived theoretical results lie in the need to find a optimal solution to the score. However, a recently proposed (Chevalley et al., 2024) algorithm show that the objective can be solved at large scale.

**Assumption 1** ($\epsilon$-Interventional Faithfulness (Chevalley et al., 2025c)). Let $\mathcal{C}$ be a structural causal model with associated DAG $\mathcal{G}$ and $\tilde{N}$ denote exogenous noise variables under interventions. We say that $(\tilde{N}, \mathcal{C})$ is *$\epsilon$-interventionally faithful* to $\mathcal{G}$ if, for every pair $i \neq j$ with $i \in \mathcal{I}$ and $j \in V$,

$$D\Big(P_{X_j}^{\mathcal{C},(\emptyset)}, P_{X_j}^{\mathcal{C},do(X_i := \tilde{N}_i)}\Big) > \epsilon \iff \text{there exists a directed path } i \rightsquigarrow j \text{ in } \mathcal{G}.$$

Unlike classical *faithfulness*, which requires every d-separation in $\mathcal{G}$ to manifest as a conditional independence in the joint distribution, $\epsilon$-interventional faithfulness imposes only a marginal shift requirement: whenever there is a directed path from an intervened node $i$ to $j$, the distribution of $X_j$ changes by at least $\epsilon$ under intervention on $i$. This weaker assumption does not rely on conditional independence structure and is preserved under marginalization, making it compatible with latent confounding. In particular, as long as interventions on observed variables induce distributional shifts that are detectable above threshold $\epsilon$, the assumption continues to hold even when hidden confounders influence both $i$ and $j$.

Chevalley et al. (2025c) show that for an intervention policy in which each intervention indicator is chosen independently as $I_k \sim \text{Bernoulli}(p_{\text{int}}), k \in V$, one can derive upper bounds on the expected number of misorientations $D_{\text{top}}(\mathcal{G}, \pi_{\text{opt}}(\mathbf{I}))$. Here $\mathbf{I} = (I_1, \ldots, I_d)$ is the random intervention vector, and $\pi_{\text{opt}}(\mathbf{I})$ denotes the optimal causal order under that intervention design, i.e., the permutation that maximizes the score function given the specific realization of $\mathbf{I}$. Writing $\pi_{\text{opt}}(\mathbf{I})$ emphasizes that the optimal order is itself a random variable induced by the random choice of interventions, and thus $D_{\text{top}}$ inherits its stochasticity through $\mathbf{I}$. In this paper, we extended those bounds for the FNR, we derive deviation bounds for both metrics, and derive all those properties for a Barabási–Albert graph distribution. A complete list of all notation is provided in Appendix B.

Although our analysis is phrased in terms of the score-optimal order $\pi_{\text{opt}}$ associated with the InterSort objective, the guarantees themselves are far more general. The quantity is simply a convenient formalization of a *possible* orientation performance achievable under the given intervention design and data assumptions. Any causal-discovery method that relies on single-node interventions and exploits the same $\epsilon$-interventional faithfulness property must obey the same upper bounds and deviation behavior, since these depend only on the structure of $\mathcal{G}$, the intervention probability $p_{int}$, and the induced reachability relations—not on the specifics of the InterSort score. Conversely, if an algorithm attains strictly higher error, our results quantify the remaining performance gap attributable to suboptimal use of the interventional information. Thus, while the InterSort score provides a clean mathematical vehicle for theoretical analysis, the deviation guarantees we derive characterize the intrinsic statistical difficulty of orientation under randomized interventions for any method operating under these assumptions.

## 4 THEORETICAL RESULTS

In what follows we assume access to both the observational distribution $P_X^{\mathcal{C},(\emptyset)}$ and the single-variable interventional distributions $P_X^{\mathcal{C}, do(X_k := \tilde{N}_k)}$ for all $k \in \mathcal{I}$, under the condition that $(\tilde{N}, \mathcal{C})$ is (*restricted*) $\epsilon$-interventionally faithful (cf. Chevalley et al. (2025c)). In the restricted setting, interventions reveal only direct parent–child relations.

**Assumption 2** (Unique optimizer). Let
$$\pi_{\text{opt}} = \arg\max_\pi^\star S\left(\pi, \epsilon, D, \mathcal{I}, P_X^{\mathcal{C},(\emptyset)}, \mathcal{P}_{\text{int}}, c\right),$$
where $\arg\max^\star$ denotes a deterministic tie-breaking rule (e.g., lexicographic ordering). Thus $\pi_{\text{opt}}$ is unique for every $(\mathbf{I}, \mathbf{E})$, and all definitions below are taken with respect to this unique optimizer.

Flipping the intervention status of a node $k$ can only affect edges that "use $k$ as evidence." An edge $(i, j)$ is already secured if $j$ is directly intervened, or if some intervened ancestor of $j$ is not an ancestor of $i$. Thus the only vulnerable edges lie in $k$'s ancestral and descendant cones. Counting these provides a structural bound on how much the error can change when $I_k$ is toggled.

**Lemma 3** (Lipschitz bound for intervention variables). *Define*
$$f(\mathbf{I}) := D_{\text{top}}\left(\mathcal{G}, \pi_{\text{opt}}(\mathbf{I})\right), \qquad g(\mathbf{I}) := \frac{f(\mathbf{I})}{|E|}.$$
*For each $k \in V$, let*
$$c_k := \max_{\mathbf{I}, \mathbf{I'}: I_\ell = I'_\ell \,\forall \ell \neq k} \left| f(\mathbf{I}) - f(\mathbf{I'}) \right|.$$
*Then*
$$c_k \leq |\text{Anc}(k)| + |\text{Desc}(k)|, \qquad c_k^g = \frac{c_k}{|E|}.$$

**Lemma 4** (Restricted case). *Under restricted $\epsilon$-interventional faithfulness, flipping $I_k$ only affects edges incident to $k$, hence*
$$c_k \leq \deg_{\text{in}}(k) + \deg_{\text{out}}(k).$$

**Lemma 5** (Edge variables). *Let*

$$f(\mathbf{I}, \mathbf{E}) := D_{\text{top}}\big(\mathcal{G}(\mathbf{E}), \pi_{\text{opt}}(\mathbf{I}, \mathbf{E})\big).$$

*If only a single edge indicator $E_{ij}$ is flipped, the set of affected edges is*

$$\mathcal{A}_{ij} := \{(k, j) \in E : i \notin \text{Pa}_{\mathcal{G}}(k)\},$$

*and the Lipschitz constant satisfies*

$$c_{ij} = |\mathcal{A}_{ij}| \le \deg_{\text{in}}(j).$$

*Remark* 6. Without Assumption 2, ties between distinct maximizers could cause $\pi_{\text{opt}}$ to jump discontinuously, leading to $c_k$ or $c_{ij}$ as large as $|E|$ and voiding any bounded-difference concentration.

The Lipschitz constants derived above translate the structural sensitivity of the error functional into a uniform control on how much any single random input (intervention or edge) can affect it. This functional property is precisely what allows us to invoke deviation inequalities such as McDiarmid's: if a function of independent random variables changes only slightly when one coordinate is altered, then the function is concentrated around its mean. Thus, the Lipschitz analysis provides the bridge from graph-theoretic properties of $\mathcal{G}$ to probabilistic concentration results for the error metrics, which form the core theoretical guarantees of this paper. In the next section, we leverage this connection to derive deviation bounds for different random graph models. Complementary results that establish upper bounds on the mean error are provided in Appendix E.

## 4.1 Setting I: Fixed Graph, Random Interventions

In this first setting, we consider the graph to be fixed, and the randomness comes only from the random choice of intervention targets.

Once we know how much a single intervention flip can change the top-order error—captured by the Lipschitz constants from the previous lemmas, McDiarmid's inequality immediately yields concentration. Normalizing by $|E|$ simply scales all constants down, so the normalized error concentrates even more tightly.

**Theorem 7** (Deviation bounds for topological errors). *Let $c := \max_{k \in V} c_k$. By McDiarmid's inequality and Lemma 3, for any $t > 0$, we have the following deviation bounds:*

**Unnormalized error:**

$$P\left(|f(\mathbf{I}) - \mathbb{E}[f(\mathbf{I})]| \ge t\right) \le 2\exp\left(-\frac{2t^2}{\sum_{k=1}^{|V|} c_k^2}\right) \le 2\exp\left(-\frac{2t^2}{|V|c^2}\right).$$

**Normalized error:**

$$P\left(|g(\mathbf{I}) - \mathbb{E}[g(\mathbf{I})]| \ge t\right) \le 2\exp\left(-\frac{2t^2|E|^2}{\sum_{k=1}^{|V|} c_k^2}\right) \le 2\exp\left(-\frac{2|E|^2 t^2}{|V|c^2}\right).$$

## 4.2 Setting II: Erdős–Rényi Graphs

In addition to the previous assumptions, we now take the underlying graph itself to be random. Specifically, we generate an Erdős–Rényi random DAG on $d$ nodes by first sampling a uniform random topological ordering $\sigma \in S_d$, and then including each forward edge $(\sigma(i), \sigma(j))$ with independent probability $p_e$ whenever $i < j$. This construction ensures acyclicity by orienting edges consistently with $\sigma$.

Formally, let $\mathbf{E} := \{E_{ij} : 1 \le i < j \le d\}$ be the set of independent edge indicators, where $E_{ij} = 1$, with i.i.d distribution Bernoulli($p_e$), denotes that the directed edge $(i, j)$ is present. The total number of edges is $|\mathbf{E}| = \sum_{1 \le i < j \le d} E_{ij}$, $\quad \mathbb{E}[|\mathbf{E}|] = p_e \frac{d(d-1)}{2}$.

We adopt the *parents-only local influence* rule, i.e. flipping an intervention $I_k$ can only affect orientations of edges directly incident to $k$ (as assumed in Section 7 of Chevalley et al. (2025c)).

Since both the intervention vector $\mathbf{I}$ and the edge set $\mathbf{E}$ are now random, the optimal ordering itself becomes random: $\pi_{\text{opt}}(\mathbf{I}, \mathbf{E})$. This notation emphasizes that the optimizer is determined jointly by the realized interventions and by the sampled random graph, and hence the topological error

$f(\mathbf{I}, \mathbf{E}) = D_{\text{top}}(\mathcal{G}(\mathbf{E}), \pi_{\text{opt}}(\mathbf{I}, \mathbf{E}))$ and its normalized form $g(\mathbf{I}, \mathbf{E}) = f(\mathbf{I}, \mathbf{E})/|\mathbf{E}|$ are random variables depending on both sources of randomness.

We now state a theorem that characterizes the deviation from the mean for the misorientation count $f$ and the false–negative rate $g = f/|E|$. The first part uses the classical bounded–differences inequality, while the second part invokes a variance bound via Bhatia–Davis together with Chebyshev's inequality, using an upper-bound on the mean derived in Lemma 20 in Appendix E.

**Theorem 8** (Deviation bounds for $f$ and $g$ in ER graphs). *Fix $d \geq 2$ and let the deterministic tie–break described in Assumption 2 select a deterministic $\pi_{\text{opt}}(\mathbf{I}, \mathbf{E})$ in the equivalence class.*

1. ***Unnormalised error.*** *There exists $C_1 > 0$ such that for every $t > 0$*

$$P\big(\big|\, f(\mathbf{I}, \mathbf{E}) - \mathbb{E}\big[f(\mathbf{I}, \mathbf{E})\big]\big| \geq t\big) \;\leq\; 2\exp\!\Big(-\frac{2t^2}{C_1\, d^4}\Big).$$

*Hence $f$ is concentrated around its mean on the $O(d^2)$ scale.*

2. ***Normalised error.*** *For any $t > 0$ and any choice of $\delta_d \in (0, 1)$,*

$$P\big(g(\mathbf{I}, \mathbf{E}) - \mathbb{E}\big[g(\mathbf{I}, \mathbf{E})\big]\big| \geq t\big) \;\leq\; \frac{1}{t^2}\left[\frac{2(1 - p_{int})^2}{(1 - \delta_d)\, p_e\, p_{int}} \cdot \frac{1}{d - 1} \;+\; \exp\!\Big(-\frac{\mu_d\, \delta_d^2}{2}\Big)\right],$$

*where $\mu_d = \mathbb{E}[|\mathbf{E}|]$. Hence $g(\mathbf{I}, \mathbf{E})$ is concentrated at rate $O\big(\frac{1}{\sqrt{d}}\big)$ around its mean.*

The theorem shows that both the raw error $f$ and the normalized error $g$ are tightly concentrated in Erdős–Rényi graphs. For $f$, deviations around the mean are exponentially unlikely once they exceed the natural $\Theta(d^2)$ scale of the number of edges. After normalization, $g$ has mean $O(1/d)$ and variance of the same order, so its fluctuations occur on the typical width $d^{-1/2}$. In other words, as the graph grows, the expectation and fluctuation of the FNR both vanishes, providing a strong guarantee of reliability for large-scale causal discovery.

In addition to the previous Erdős–Rényi setting, we now consider the sparse regime, where the edge probability scales inversely with the number of nodes, $p_e := \frac{c}{d}$, $c > 0$. In this regime, the expected number of edges is $\mathbb{E}[|E|] = \Theta(d)$, so each node has constant expected degree. Thus the graph remains sparse even as $d$ grows, in contrast to the dense case above. This leads to an upper-bound for the mean that is constant in $d$ (Lemma 22). As a result, variance-based arguments such as Chebyshev are not sufficient, and establishing deviation bounds requires stronger concentration tools.

We next state a theorem that characterizes the deviation of the misorientation count $f$ and the normalized error $g = f/|E|$ from their expectations in this sparse regime. The proof relies on the *typical bounded differences theorem* for functions of 0–1 variables (Warnke, 2016) (see Theorem 13 in Appendix C), which sharpens concentration once atypical high-degree configurations are excluded. Specifically, standard results show that the maximum degree of an Erdős–Rényi DAG in this regime is $O(\log d)$ with high probability, which provides the required Lipschitz control for our analysis.

**Theorem 9** (Deviation bounds in the sparse regime $p_e = c/d$). *Fix $c > 0$ and let $K > 0$. For all $d \geq 2$ the following hold:*

1. ***Unnormalised topological error.*** *For every $t > 0$,*

$$P\big(\big|\, f(\mathbf{I}, \mathbf{E}) - \mathbb{E}\big[f(\mathbf{I}, \mathbf{E})\big]\big| \geq t\big) \;\leq\; \underbrace{d^{-K}}_{\substack{\text{bounding \#edges and } \deg_{\max}}} + 2\exp\!\Big(-\frac{t^2}{2c_1\, d\log^2 d + \frac{2}{3}c_2 t\log d}\Big)$$

*with universal constants $c_1, c_2 > 0$. Consequently $f(\mathbf{I}, \mathbf{E})$ is concentrated at rate $O\big(\sqrt{d}\,\log d\big)$ around its mean.*

2. ***Normalised topological error.*** *For every $t > 0$,*

$$P\big(\big|\, g(\mathbf{I}, \mathbf{E}) - \mathbb{E}\big[g(\mathbf{I}, \mathbf{E})\big]\big| \geq t\big) \;\leq\; \underbrace{d^{-K}}_{\substack{\text{bounding \#edges and } \deg_{\max}}} + 2\exp\!\Big(-\frac{d\, t^2}{c_3\log^2 d + \frac{2}{3}c_4 t\log d}\Big)$$

*with universal constants $c_3, c_4 > 0$. Hence $g(\mathbf{I}, \mathbf{E})$ is concentrated at rate $O\big(\frac{\log d}{\sqrt{d}}\big)$ around its mean.*

Compared to the dense Erdős–Rényi case (where $p_e = \Theta(1)$ and $g$ concentrates with typical width $O(d^{-1/2})$ around a vanishing mean $\Theta(1/d)$), the sparse regime $p_e = c/d$ behaves differently: the mean of $g$ does not vanish (Lemma 22), yet $g$ still concentrates with typical width $O\left(\frac{\log d}{\sqrt{d}}\right)$. This is weaker than the dense case by only a logarithmic factor. The $\log d$ penalty arises from controlling the maximum degree ($\deg_{\max} = O(\log d)$ w.h.p.); once such rare high-degree events are excluded, Warnke's typical bounded–differences inequality yields the stated sub-Gaussian tail. For the unnormalized error $f$, fluctuations are $O(\sqrt{d}\log d)$ around an $O(d)$ mean, again reflecting the $\log d$ overhead due to maximum–degree control in the sparse regime.

## 4.3 Setting III: Generalized Barabási–Albert Graphs

Many real-world networks—from biological systems to social interactions—exhibit heterogeneous connectivity patterns, often characterized by a small number of highly connected hubs and many nodes with few connections. To capture such structures, we extend our analysis to a generalized version of the Barabási–Albert (BA) (Barabási & Albert, 1999) model that incorporates an *initial attractiveness* parameter $\kappa > 0$ (Dorogovtsev et al., 2000).

In this generalized model, one begins with a small seed network of $m_0$ nodes. New nodes are introduced sequentially, and each new node attaches to $m$ existing nodes. The probability that a new node connects to an existing node $i$ is given by

$$P(i) = \frac{k_i + \kappa}{\sum_j (k_j + \kappa)},$$

where $k_i$ is the current degree of node $i$. This leads to a power-law degree distribution (Bollobás et al., 2001; Dorogovtsev et al., 2000; Buckley & Osthus, 2004) of the form

$$P(k) \sim k^{-\gamma},$$

with an exponent $\gamma$ that depends on both the number of links $m$ per new node and the attractiveness parameter $\kappa$. By tuning $\kappa$, the model can generate networks with a range of exponents, thereby accommodating different levels of heterogeneity observed in empirical data.

Since our analysis concerns causal discovery in directed acyclic graphs (DAGs), we must ensure that the network generated by this model is both directed and acyclic. To achieve this, we impose a natural time-ordering on the nodes based on their arrival sequence. Specifically, we direct all edges from older nodes (those that appeared earlier in the growth process) to younger nodes (those added later). This rule guarantees that no cycles can form, as it is impossible to have a directed edge pointing from a newer node back to an older one. With this construction, the generalized BA model produces a directed acyclic graph (DAG) that retains the desired scale-free properties.

Crucially, the emergence of hub nodes can have a pronounced effect on the concentration of topological errors in causal discovery. In the sections that follow, we derive deviation bounds for these errors in the context of generalized BA graphs and discuss how varying $\gamma$ impacts the robustness of causal inference methods.

If we start our generating process from an empty graph, after $d$ steps, we will obtain a DAG of $d$ nodes and $|E| = d \cdot m$ edges.

**Lemma 10** (High-Probability Bound on Node Degrees). *Let $\mathcal{G}$ be a generalized Barabási–Albert DAG with $d$ nodes constructed as described above. Then there exist a constant $C > 0$ such that, with high probability, every node $i \in V$ satisfies*

$$in\text{-}deg(i) + out\text{-}deg(i) \leq m + C\,d^\beta,$$

*where $\beta = \frac{1}{\gamma-1}$ and $\gamma = 2 + \frac{\kappa}{m}$. We denote this event $\mathcal{E}_{BA}$, with support $E_{BA}$*

Lemma 10 is a direct corollary of several well-known results on initial–attractiveness preferential-attachment processes. Van Der Hofstad (2024, Theorem 8.14) prove that for any fixed $(m, \kappa)$ the rescaled maximum degree $d^{-\beta}\Delta_d$ converges almost surely to a non–degenerate random limit; almost-sure convergence immediately implies the stated *with-high-probability* bound. See also seminal results such as (Móri, 2005; Buckley & Osthus, 2004; Dorogovtsev et al., 2000; Bollobás & Riordan, 2003; Bollobás et al., 2001).

Preferential-attachment graphs have heavy-tailed degrees with a well-understood maximum degree. Under the parents-only assumption, flipping one intervention can only affect edges incident to that node, so the per-coordinate sensitivity is at most $m + C\,d^\beta$. Applying McDiarmid to the intervention vector then gives the deviations.

**Theorem 11** (Deviation Bounds in Generalized BA Graphs). *Let $\mathcal{G}$ be a generalized Barabási–Albert DAG on $d$ nodes, with parameters $m > 0$, $\kappa > 0$, and $\gamma = 2 + \frac{\kappa}{m}$. Under the natural time-ordering of edges and with $|E| = m\,d$ (starting from an empty seed), there exist a constants $C > 0$ such that, with high probability, flipping one node's intervention indicator changes $f(\mathbf{I})$ by at most $m + C\,d^\beta$,*

*Consequently, with high probability, by McDiarmid's inequality:*

***Unnormalized error.*** *For any $t > 0$, $\mathbf{E} \in E_{BA}$,*

$$P\Big(\,\big|f(\mathbf{I}) - \mathbb{E}[f(\mathbf{I})|\mathbf{E}]\big|\,\geq t\Big|\mathbf{E}\Big) \;\leq\; 2\exp\!\Big(-\frac{2\,t^2}{d\,\big(m + C\,d^\beta\big)^2}\Big).$$

*In other words, $f(\mathbf{I})$ concentrates around its conditional mean up to deviations on the scale $d^{\beta + \frac{1}{2}}$ for any fixed graph in the high probability event.*

***Normalized error.*** *For $g(\mathbf{I}) = f(\mathbf{I})\big/|E| = f(\mathbf{I})/(m\,d)$, flipping an intervention of any single node changes $g(\mathbf{I})$ by at most $(m + C\,d^\beta)/(m\,d)$. Hence, letting $t > 0$, $\forall \mathbf{E} \in E_{BA}$,*

$$P\Big(\,\big|g(\mathbf{I}) - \mathbb{E}[g(\mathbf{I})|\mathbf{E}]\big|\,\geq t\Big|\mathbf{E}\Big) \;\leq\; 2\exp\!\Big(-\frac{2\,t^2}{\sum_{i=1}^{d}\big(\frac{m+C\,d^\beta}{m\,d}\big)^2}\Big).$$

*Since $\sum_{i=1}^{d}\big(\frac{m+C\,d^\beta}{m\,d}\big)^2 = O\big(d^{2\beta-1}\big)$, the exponent scales like $-\frac{2\,t^2}{d^{2\beta-1}}$. Consequently, $g(\mathbf{I})$ concentrates around $\mathbb{E}[g(\mathbf{I})|\mathbf{E}]$ at the $d^{\beta-\frac{1}{2}}$ scale if $2\beta > 1$, or even faster when $\gamma > 3$ (i.e. $\beta < \frac{1}{2}$), for any fixed graph in the high probability event $\mathcal{E}_{BA}$.*

## 5 EMPIRICAL ILLUSTRATION OF DEVIATION CONCENTRATION

Our theoretical results establish non-asymptotic deviation bounds for the normalized topological error $g(\mathbf{I}, \mathbf{E}) = D_{\text{top}}(\mathcal{G}(\mathbf{E}), \pi_{\text{opt}}(\mathbf{I}, \mathbf{E}))/|E|$. These guarantees concern the *optimal solution* to the Intersort score, for which no polynomial-time algorithm is known. As such, any practical simulation must rely on an approximation algorithm, which may introduce bias. We therefore emphasize that the following experiments are provided only as an illustration of deviation concentration trends, and not as a direct test of our finite-sample bounds. We employ the differentiable relaxation DIFFINTERSORT (Chevalley et al., 2024), which approximates the maximizer of the score using Sinkhorn-based optimization. While DiffIntersort is not guaranteed to recover the true optimum, prior work shows that it achieves close approximations in practice and is scalable up to 2000 variables, making it suitable for empirical illustration of the scaling behavior. We generated synthetic random DAGs from three graph families: (i) Erdős–Rényi (ER) with edge probability $p_e \in \{0.2, 0.4, 0.6\}$, (ii) scale-free DAGs via preferential attachment with hub parameter $\kappa \in \{1.0, 3.0, 9.0\}$ and $m = 3$ edges per variable, and (iii) scale-free ER graphs with average degree per node $c \in \{2, 3, 5\}$. Graph sizes ranged from $d = 30$ up to $d = 2000$. For each graph, we sampled single-variable intervention vectors with probability $p_{\text{int}} \in \{0.25, 0.5, 0.75\}$. For each configuration we repeated 10 runs and report the mean and deviation width of $g(\mathbf{I}, \mathbf{E})$. To match the theoretical focus on concentration, we report the *typical deviation width* of the empirical mean of the normalized topological, measured by the interquartile range (IQR). Standard deviations are reported in the appendix (Figure 2).

**Results.** Figure 1 reports the empirical deviation width of $g(\mathbf{I}, \mathbf{E})$ as a function of $d$ for representative parameter settings. Across all graph families, variability decreases with dimension, in line with our concentration theorems. For ER and sparse ER graphs, the simulations clearly confirm the theory, showing vanishing deviation as $d$ grows. For scale-free BA graphs, the theoretical predictions are more nuanced: when $\kappa = 9.0$, the deviation should vanish; for $\kappa = 3.0$, it should remain constant; and for $\kappa = 1.0$, it should grow at rate $O(d^{1/4})$. Empirically, the deviation shrinks for $\kappa = 3.0$ and $\kappa = 9.0$, while for $\kappa = 1.0$ it remains less stable and fluctuates more strongly, consistent with the weaker theoretical control in this regime. Overall, the simulations provide strong support for the theoretical results.

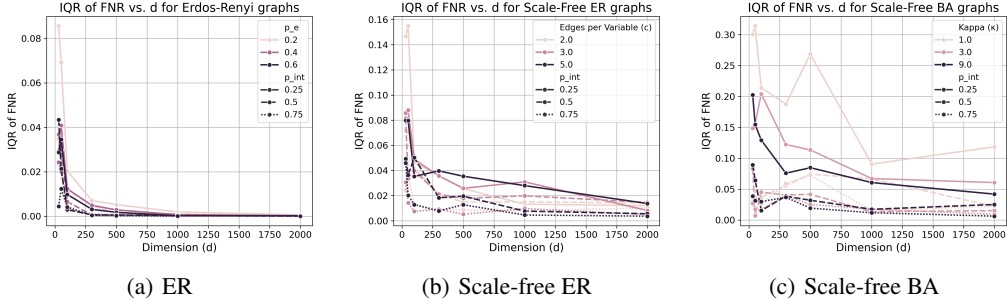

(a) ER        (b) Scale-free ER        (c) Scale-free BA

Figure 1: Interquartile range (IQR) of the FNR as a function of graph size $d$. For each graph family, results are shown across three density parameters and three values of intervention coverage $p_{\text{int}}$. The IQR decreases with $d$, demonstrating vanishing variability as predicted by our theoretical results, except for scale-free BA graphs with $\kappa = 1$, which correspond to the heavy-tailed regime with exponent $\gamma = \frac{7}{3} < 3$.

## 6    LIMITATIONS

Our analysis relies on several simplifying assumptions. $\epsilon$-interventional faithfulness still requires that interventions induce measurable shifts in downstream distributions, which may fail in high-noise systems or with very weak effects. The random graph models we study are stylized abstractions and do not capture domain-specific structure such as modularity or hierarchy. Nonetheless, Erdős–Rényi and generalized Barabási–Albert distributions reflect several key structural features found in real-world networks, such as sparsity, variation in local neighborhood sizes, and heavy-tailed degree patterns, even though no single model captures all biological or neural properties simultaneously. Likewise, our assumption of random single-node interventions represents a conservative, worst-case setting: more targeted or optimized intervention designs can only improve the guarantees. Our assumption of access to a sufficiently large number of interventions is not always realistic, but it is increasingly common in experimental settings such as high-throughput perturbation screens (Replogle et al., 2022) or large-scale neurostimulation studies (Peña-Gómez et al., 2012; Yang et al., 2021; Momi et al., 2025). We note that our deviation bounds do not depend on the intervention probability $p_{int}$, a limitation inherent to McDiarmid-style inequalities; exploring proof techniques that yield deviation-level dependence on $p_{int}$ is an interesting direction for future work. We also assume that causal systems are acyclic, excluding feedback loops that arise in many dynamical settings. Finally, our empirical evaluation uses DiffIntersort as a proxy search algorithm. Our theoretical results concern the optimum of the score and not the performance of any specific approximation algorithm; improving practical algorithms for large-scale settings is complementary future work.

## 7    CONCLUSION

We presented finite-dimension deviation bounds on the false-negative rate of causal discovery under single-variable random interventions and an $\epsilon$-interventional faithfulness assumption that accommodates latent confounding. Our results demonstrate that graphs exhibiting structural regimes commonly observed in empirical systems have desirable statistical regularity: the false-negative rate not only remains bounded but concentrates tightly, and in many regimes, vanishes asymptotically. These findings challenge worst-case intuitions and suggest that high-dimensional structure learning can, under plausible conditions, be more stable than previously expected. Our results offer a principled way to reason about the trade-off between intervention budget and error, providing guidance for designing large-scale interventional studies with controlled orientation error rates. Future work may extend these guarantees to adaptive intervention policies, alternative error metrics, and more general graph distributions, paving the way for robust and efficient causal inference in large scale complex systems.

## 8 REPRODUCIBILITY STATEMENT

Complete proofs for all the theoretical results are provided in Appendix F, with corresponding notations in Appendix B. For the simulations, the parameters used are described in Section 5, and corresponding code is provided as supplementary material.

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

## A EXTENDED RELATED WORK

Theoretical guarantees in interventional causal discovery often focus on deriving upper and lower bounds on the number of interventions required to fully recover a causal DAG. Given the the observational MEC, the goal is to use targeted interventions to oriented the unoriented edges in the MEC. Intervention sets can be atomic, where only one variable can be intervened on per experiments, or multiple, where the set of intervened variables can be up to $d$. Early work showed that for multiple intervention sets, $\mathcal{O}(\log d)$ intervention sets are sufficient, and in the worst-case necessary to identify the true graph (Eberhardt et al., 2005), and that for atomic sets, $d - 1$ experiments are sufficient (Eberhardt et al., 2006). Those result were later refined to show that $\mathcal{O}(\log \omega(\mathcal{G}))$ (Hauser & Bühlmann, 2014), where $\omega(\mathcal{G})$ is the maximum clique size of the graph, are actually sufficient for multiple interventions, as was conjecture by Eberhardt (2008). These worst-case analyses take an adversarial view of the problem, where the goal is then to find an experiment selection that is robust to any adversary. Eberhardt (2010) expands on this game-theoretic view and proposes to use randomization to improve the expected worst case, showing that for atomic intervention $\Theta(d)$ experiments are sufficient. Hu et al. (2014) used this idea to show that for multiple interventions, a randomized algorithm needs $\mathcal{O}(\log \log d)$ experiment in the worse case in expectation. Similar mini-max bounds were derived under various settings (Shanmugam et al., 2015; Kocaoglu et al., 2017a). Lower bounds on the minimum number of needed experiments to orient any DAG in a MEC has also been analyzed (Squires et al., 2020a; Porwal et al., 2022). Relaxing the causal sufficiency

assumption, Addanki et al. (2020); Kocaoglu et al. (2017b) derive sufficient set of interventions to recover the graph, as well has its latent variables. Regarding "average case" scenarios, for example by considering a parameterized graph distribution, the literature is more limited. For example, Hu et al. (2014); Katz et al. (2019) show that for an Erdős-Rényi graph distribution, the sufficient number of interventions to recover the graph is constant in expectation. Average case analysis may provide more practical insights for real-world causal discovery compared to worse-case analysis. Existing literature is also limited in the sense that faithfulness or knowledge of the observational MEC, as well as causal sufficiency, are often assumed.

An adjacent line of research considers the problem of intervention design under budget constraints (e.g. limited number of interventions), or adaptive sequential selection of interventions (He & Geng, 2008; Hyttinen et al., 2013; Ghassami et al., 2018; Hauser & Bühlmann, 2014; Sussex et al., 2021; Agrawal et al., 2019). Here, the goal is to maximize the utility of the limited set of performed interventions. Given that finding the optimal set is in general not tractable, the focus is around finding provably approximate (greedy) algorithms.

The characterization of necessary intervention sets to *fully* identify causal graphs also does not answer the question of partial identification under a limited of interventions. In seminal work, Hauser & Bühlmann (2012) characterize an interventional Markov equivalence class ($\mathcal{I}$-MEC) given multiple interventions and propose a greedy algorithm for structure learning with interventional data. $\mathcal{I}$-MECs can only be smaller than the corresponding observational MEC, and thus interventions improve identifiability. Many work followed and refined on this idea, both around characterizing equivalences classes under various assumptions (Yang et al., 2018; Jaber et al., 2020; Kocaoglu et al., 2019; Squires et al., 2020b) as well as proposing more efficient learning algorithms (Brouillard et al., 2020; Ke et al., 2019; 2023; Mooij et al., 2020; Nazaret et al., 2023; Wang et al., 2017; Faria et al., 2022; Lorch et al., 2022).

## B    LIST OF NOTATIONS

$V$  The set of nodes (or random variables), with $V = \{1, 2, \ldots, d\}$.

$d$  The number of nodes (or dimensionality) in the causal graph.

$\mathcal{G} = (V, E)$  A directed acyclic graph (DAG) representing the causal structure, where $E \subseteq V \times V$ is the set of directed edges.

$\mathbf{A}^{\mathcal{G}}$  The adjacency matrix of $\mathcal{G}$, where

$$\mathbf{A}^{\mathcal{G}}_{ij} = \begin{cases} 1, & \text{if } (i, j) \in E, \\ 0, & \text{otherwise.} \end{cases}$$

$\mathrm{Pa}(j)$  The set of parents of node $j$, i.e., $\mathrm{Pa}(j) = \{i \in V \mid (i, j) \in E\}$.

$\mathrm{An}_{\mathcal{G}}(j)$  The set of ancestors of node $j$ (all nodes with directed paths leading to $j$).

$\mathrm{De}_{\mathcal{G}}(j)$  The set of descendants of node $j$ (all nodes reachable by directed paths from $j$).

$\mathcal{C} = (\mathbf{S}, P_N)$  A structural causal model (SCM), where $\mathbf{S}$ is the set of structural equations and $P_N$ is the joint distribution over the exogenous noise variables $N = (N_1, \ldots, N_d)$.

$X_j$  The $j$-th random variable in the SCM, with structural equation

$$X_j = f_j\big(\mathbf{X}_{\mathrm{Pa}(j)}, N_j\big).$$

$N_j$  The exogenous noise variable associated with $X_j$.

$\tilde{N}_k$  The new exogenous noise variable used in an intervention on $X_k$.

$\mathcal{I}$  The set of intervention targets, where $\mathcal{I} \subseteq V$.

$\mathbf{I} = (I_1, \ldots, I_d)$  The intervention indicator vector, where

$$I_k = \begin{cases} 1, & \text{if } k \in \mathcal{I}, \\ 0, & \text{otherwise.} \end{cases}$$

$P_X^{\mathcal{C}, (\emptyset)}$  The observational distribution of $X$ corresponding to the SCM $\mathcal{C}$ (i.e., with no interventions).

$P_X^{\mathcal{C},do(X_k:=\tilde{N}_k)}$ The interventional distribution of $X$ when the structural equation for $X_k$ is replaced by $X_k = \tilde{N}_k$.

$\pi$ A permutation of $V$ representing a candidate causal order.

$\Pi^*$ The set of all causal orders (permutations) consistent with $\mathcal{G}$.

$D_{\mathbf{top}}(\mathcal{G}, \pi)$ The *topological error* of ordering $\pi$ with respect to $\mathcal{G}$, defined as

$$D_{\text{top}}(\mathcal{G}, \pi) = \sum_{\pi(i) > \pi(j)} \mathbf{A}_{ij}^{\mathcal{G}}.$$

$D_{\mathbf{top}}(\mathcal{G}, \pi_{\text{opt}}(\mathbf{I}))$ The topological error of the optimal order under a fixed graph $\mathcal{G}$ when the intervention vector $\mathbf{I}$ is random. Here $\pi_{\text{opt}}(\mathbf{I})$ denotes the score-maximizing permutation given the specific realization of $\mathbf{I}$. Thus $D_{\text{top}}(\mathcal{G}, \pi_{\text{opt}}(\mathbf{I}))$ is a random variable through its dependence on $\mathbf{I}$, while the graph $\mathcal{G}$ is treated as fixed.

$D_{\mathbf{top}}(\mathcal{G}, \pi_{\text{opt}}(\mathbf{I}))$ The topological error of the optimal causal order when the underlying graph $\mathcal{G}$ is fixed and only the intervention vector $\mathbf{I}$ is random. Here $\pi_{\text{opt}}(\mathbf{I})$ denotes the order that maximizes the score function given the specific realization of $\mathbf{I}$. Thus, $D_{\text{top}}$ is a random variable induced solely by the randomness in the intervention design.

$D$ A statistical distance (or divergence) function

$$D \colon \mathcal{P}(\mathcal{M}) \times \mathcal{P}(\mathcal{M}) \to [0, \infty),$$

which measures the discrepancy between two probability distributions.

$\epsilon$ A significance threshold used in the definition of $\epsilon$-interventional faithfulness and in the score function.

$c$ A constant with $c > \epsilon$, used in scaling terms of the score function.

$S(\pi, \epsilon, D, \mathcal{I}, P_X^{\mathcal{C},(\emptyset)}, \mathcal{P}_{\text{int}}, c)$ The score function defined in equation 2 that evaluates how well a candidate ordering $\pi$ aligns with the interventional data.

$f(\mathbf{I})$ The unnormalized topological error in the fixed-graph setting, defined as

$$f(\mathbf{I}) := D_{\text{top}}\big(\mathcal{G}, \pi_{\text{opt}}(\mathbf{I})\big).$$

This quantity is random only through the intervention vector $\mathbf{I}$.

$g(\mathbf{I})$ The normalized topological error (false negative rate, FNR) in the fixed-graph setting, defined as

$$g(\mathbf{I}) := \frac{f(\mathbf{I})}{|E|},$$

where $|E|$ is the total number of edges in the fixed graph $\mathcal{G}$.

$f(\mathbf{I}, \mathbf{E})$ The unnormalized topological error, defined as

$$f(\mathbf{I}, \mathbf{E}) := D_{\text{top}}\big(\mathcal{G}(\mathbf{E}), \pi_{\text{opt}}(\mathbf{I}, \mathbf{E})\big).$$

Here the dependence on $(\mathbf{I}, \mathbf{E})$ is made explicit, since both the optimal order and hence the error are random variables determined by the intervention vector and the random graph.

$g(\mathbf{I}, \mathbf{E})$ The normalized topological error (false negative rate, FNR), defined as

$$g(\mathbf{I}, \mathbf{E}) := \frac{f(\mathbf{I}, \mathbf{E})}{|\mathbf{E}|},$$

where $|\mathbf{E}|$ is the total number of edges in $\mathcal{G}(\mathbf{E})$.

$c_k$ The Lipschitz constant measuring the maximum change in $f(\mathbf{I}, \mathbf{E})$ upon flipping the $k$-th intervention indicator, i.e.,

$$c_k := \max_{\substack{\mathbf{I}, \mathbf{I}' \in \{0,1\}^d \\ I_\ell = I'_\ell \, \forall \, \ell \neq k}} \Big| f(\mathbf{I}, \mathbf{E}) - f(\mathbf{I}', \mathbf{E}) \Big|.$$

## C    BOUNDED DIFFERENCES THEOREMS

**Theorem 12** (Bounded differences inequality, McDiarmid et al. (1989)). *Let $X = (X_1, \dots, X_N)$ be independent random variables with $X_k$ taking values in $\Lambda_k$. Suppose $f : \prod_{k=1}^N \Lambda_k \to \mathbb{R}$ satisfies*

$$|f(x) - f(x')| \leq c_k \quad \text{whenever } x, x' \text{ differ only in the } k\text{th coordinate.} \tag{3}$$

*Let $\mu = \mathbb{E}[f(X)]$. Then for all $t \geq 0$,*

$$\mathbb{P}\big(f(X) \geq \mu + t\big) \leq \exp\Big(-\frac{2t^2}{\sum_{k=1}^N c_k^2}\Big).$$

*An analogous bound holds for the lower tail $\mathbb{P}(f(X) \leq \mu - t)$.*

**Theorem 13** (Typical bounded differences inequality, Warnke (2016)). *Let $X = (X_1, \dots, X_N)$ be independent with $X_k \in \Lambda_k$, and let $\Gamma \subseteq \prod \Lambda_k$ be an event. Suppose $f : \prod \Lambda_k \to \mathbb{R}$ satisfies*

$$|f(x) - f(x')| \leq \begin{cases} c_k, & x \in \Gamma, \\ d_k, & x \notin \Gamma, \end{cases}$$

*whenever $x, x'$ differ only in coordinate $k$. For any $\gamma_k \in (0, 1]$, let $e_k = \gamma_k(d_k - c_k)$. Then there is a "bad" event $\mathcal{B}$ with*

$$\mathbb{P}(\mathcal{B}) \leq \sum_{k=1}^N \frac{\mathbb{P}(\neg\Gamma)}{\gamma_k},$$

*and for all $t \geq 0$, on $\neg\mathcal{B}$ one has*

$$\mathbb{P}\big(f(X) \geq \mathbb{E}f(X) + t \text{ and } \neg\mathcal{B}\big) \leq \exp\Big(-\frac{t^2}{2\sum_{k=1}^N (c_k + e_k)^2}\Big).$$

**Theorem 14** (Typical bounded differences for 0–1 variables, Warnke (2016)). *Let $X = (X_1, \dots, X_N)$ be independent with $X_k \in \{0, 1\}$ and $p_k = \mathbb{P}(X_k = 1)$. Let $\Gamma \subseteq \{0, 1\}^N$, and suppose $f : \{0, 1\}^N \to \mathbb{R}$ satisfies the same one–sided Lipschitz condition above with coefficients $c_k, d_k$ and compensation $e_k = \gamma_k(d_k - c_k)$. Define $C = \max_k(c_k + e_k)$ and $V = \sum_k (1 - p_k) p_k (c_k + e_k)^2$. Then there is a "bad" event $\mathcal{B}$ with the same bound as above, and for all $t \geq 0$,*

$$\mathbb{P}\big(f(X) \geq \mathbb{E}f(X) + t \text{ and } \neg\mathcal{B}\big) \leq \exp\Big(-\frac{t^2}{2V + 2Ct/3}\Big).$$

## D    THEOREMS FROM CHEVALLEY ET AL. (2025C)

**Lemma 15** (Sufficient condition for orienting an edge Chevalley et al. (2025c)). *Assume that we are given $P_X^{\mathcal{C}, (\emptyset)}$ and $P_X^{\mathcal{C}, do(X_k := \tilde{N}_k)}, \forall k \in \mathcal{I}$, such that $(\tilde{N}, \mathcal{C})$ is $\epsilon$-interventionally faithful for some $\epsilon > 0$, and let $\pi_{opt} \in \operatorname{argmax}_\pi \mathcal{S}(\pi)$. Let $(i, j) \in E$, then if $j \in \mathcal{I}$ or for some $k \in AN_j^{\mathcal{G}} \backslash AN_i^{\mathcal{G}}, k \in \mathcal{I}$, then $\pi_{opt}(i) < \pi_{opt}(j)$.*

**Lemma 16** (Sufficient condition for orienting an edge under restricted interventional faithfulness Chevalley et al. (2025c)). *Let $(i, j) \in E$, then if $j \in \mathcal{I}$ or for some $k \in Pa_j^{\mathcal{G}} \setminus Pa_i^{\mathcal{G}}, k \in \mathcal{I}$, then $\pi_{opt}(i) < \pi_{opt}(j)$.*

## E    ADDITIONAL THEORETICAL RESULTS

We here derive a set of theoretical results that upper-bound the expected FNR for all the graph types considered in this paper. Corresponding simulations to verify the theory can be found in Appendix G.2.

### E.1 BOUNDS ON THE MEAN FOR FIXED GRAPH, RANDOM INTERVENTIONS

An edge $(i, j)$ fails to orient correctly only if neither $j$ nor any of the "useful" ancestors of (those not shared with $i$) are intervened. Under independent random interventions, this failure probability is a simple power of $(1 - p_{int})$, and summing over edges gives the expected FNR. A direct Markov inequality then converts this expectation into a tail bound.

**Lemma 17** (Probability bound on the fraction of topological errors). *Let $c_e \in \mathbb{R}$, $0 < c_e < 1$. and the graph is non-empty, i.e. $|\mathcal{G}| = |E| > 0$. Then we have $\forall c_e$*

$$P\left(g(\mathbf{I}) \geq c_e\right) \leq \frac{1}{c_e |E|} \sum_{(i,j) \in \mathcal{G}} (1 - p_{int})^{|\mathbf{AN}_j^{\mathcal{G}} \cup \{j\} \setminus \mathbf{AN}_i^{\mathcal{G}}|}$$

*The probability is also strictly bounded by* 1.

**Corollary 18** (Expected FNR). *Under the same assumptions, we have*

$$\mathbb{E}\left[g(\mathbf{I})\right] \leq \frac{1}{|E|} \sum_{(i,j) \in \mathcal{G}} (1 - p_{int})^{|\mathbf{AN}_j^{\mathcal{G}} \cup \{j\} \setminus \mathbf{AN}_i^{\mathcal{G}}|}$$

### E.2 BOUNDS ON THE MEAN FOR ER GRAPHS

In an Erdős–Rényi DAG, the expected number of misorientations grows only linearly with $d$, whereas the number of edges is quadratic. We therefore expect the normalized error to be small with high probability. Formally, we pair a Markov bound on the numerator with a Chernoff lower tail on the random denominator $|E|$, obtained by conditioning on the event that $|E|$ is close to its mean.

**Lemma 19** (Probability bound for FNR). *For any $0 < c_e < 1$ and $0 < \delta < 1$,*

$$P\left(g(\mathbf{I}, \mathbf{E}) \geq c_e\right) \leq \frac{2(1 - p_{int})^2}{c_e(1 - \delta) p_{int} p_e (d - 1)} + \exp\left(-\frac{\delta^2 p_e d(d - 1)}{4}\right).$$

The proof splits on a "good" event where $|E|$ is not too small. On that event, the ratio is small; off it, the event's probability is exponentially tiny. This yields an explicit finite-$d$ tail bound for the FNR in ER graphs.

Because $\mathbb{E}[D_{top}] = O(d)$ while $\mathbb{E}[|E|] = \Theta(d^2)$ in ER graphs, the expected FNR should decay to zero. The only technicality is that $|E|$ is random; again a good-event/bad-event split resolves this.

**Lemma 20** (Fraction of Misoriented Edges Vanishes in Expectation). *Let $\mathcal{G}$ be an Erdős–Rényi DAG on $[d]$ with edge inclusion probability $p_e \in (0, 1]$, and let $p_{int} \in (0, 1]$. For any sequence $\delta_d \in (0, 1)$, define the event*

$$A_{\delta_d} := \left\{ |E| \geq (1 - \delta_d) \mu_d \right\}, \qquad \mu_d := \mathbb{E}[|E|] = \frac{p_e}{2} d(d - 1).$$

*Then*

$$\mathbb{E}\left[\frac{D_{\text{top}}(\mathcal{G}, \pi_{\text{opt}})}{|E|}\right] \leq \frac{2(1 - p_{int})^2}{(1 - \delta_d) p_e p_{int}} \cdot \frac{1}{d - 1} + \exp\left(-\frac{\mu_d \delta_d^2}{2}\right).$$

*In particular, choosing any $\delta_d \downarrow 0$ (e.g. $\delta_d = d^{-1/2}$) yields*

$$\mathbb{E}\left[\frac{D_{\text{top}}(\mathcal{G}, \pi_{\text{opt}})}{|E|}\right] \leq \frac{2(1 + o(1))(1 - p_{int})^2}{p_e p_{int}} \cdot \frac{1}{d} + \exp\left(-\Omega(d)\right),$$

*and hence $\lim_{d \to \infty} \mathbb{E}[D_{\text{top}}(\mathcal{G}, \pi_{\text{opt}})/|E|] = 0$.*

On the good event $|E| \sim d^2$, the ratio is $O(\frac{1}{d})$; on the bad event the ratio is at most 1 but the event is exponentially rare. Taking expectations yields the vanishing limit.

### E.3 BOUND ON THE MEAN FOR SCALE-FREE ER GRAPHS

When the graph is sparse with $p_e := \frac{c}{d}$, the number of edges is only $O(d)$, so normalization is weaker and concentration should be less sharp. Nevertheless, the same Markov+Chernoff recipe delivers explicit non-asymptotic FNR bounds, and one can read the precise dependence on $c$ and $p_{int}$.

**Lemma 21** (Probability Bounds for FNR in a Scale-Free Setting). *For any $0 < c_e < 1, 0 < \delta < 1$, $c_e, \delta \in \mathbb{R}$,*

$$P\left(g(\mathbf{I}, \mathbf{E}) \geq c_e\right) \leq \frac{2\left(1 - p_{int}\right)^2}{(1 - \delta)c_e \cdot c \cdot p_{int}} + \exp\left(-\frac{\delta^2 cd}{4}\right).$$

*We have at infinity:*

$$\lim_{d \to \infty} P\left(g(\mathbf{I}, \mathbf{E}) \geq c_e\right) \leq \frac{2(1 - p_{int})^2}{c_e c\, p_{int}}\left[1 - \frac{1}{p_{int} \cdot c}\left(1 - e^{-p_{int} \cdot c}\right)\right]$$

The proof mirrors the ER case after substituting $\mathbb{E}[|E|] = \Theta(d)$. For the asymptotic expression, we use the known limit for $\mathbb{E}[D_{top}]/d$ (Chevalley et al., 2025c).

**Lemma 22** (Expectation Bound for FNR in a Scale-Free Setting). *For any sequence $\delta_d \in (0, 1)$, we have*

$$\mathbb{E}\left[g(\mathbf{I}, \mathbf{E})\right] = \mathbb{E}\left[\frac{D_{\text{top}}(\mathcal{G}, \pi_{\text{opt}})}{|E|}\right] \leq \frac{2(1 - p_{int})^2}{(1 - \delta_d)\, c\, p_{int}} \cdot \frac{d}{d - 1} + \exp\left(-\frac{\mu_d\, \delta_d^2}{2}\right).$$

*In particular, taking any $\delta_d \downarrow 0$ with $\mu_d \delta_d^2 \to \infty$ (e.g. $\delta_d = d^{-1/2}$) yields*

$$\mathbb{E}\left[g(\mathbf{I}, \mathbf{E})\right] \leq \frac{2(1 + o(1))(1 - p_{int})^2}{c\, p_{int}} + \exp\left(-\Omega(d)\right).$$

**Corollary 23** (Asymptotic expectation bound). *Under the same assumptions, we have*

$$\limsup_{d \to \infty} \mathbb{E}\left[\frac{D_{\text{top}}(\mathcal{G}, \pi_{\text{opt}})}{|E|}\right] \leq \frac{2(1 - p_{int})^2}{c\, p_{int}}\left[1 - \frac{1}{p_{int}\, c}\left(1 - e^{-p_{int} c}\right)\right].$$

### E.4 BOUNDS ON THE MEAN FOR SCALE-FREE BA GRAPHS

**Theorem 24.** *Let $\mathcal{I}$ be chosen uniformly at random, $\mathcal{G}$ be a random generalized Barabási–Albert model directed acyclic graph with $m_0 = 0$ and $m > 0$, where $p_{int} := P(i \in \mathcal{I})\forall i \in V, 0 < p_{int} < 1$, then $\mathbb{E}[D_{top}(\mathcal{G}, \pi_{opt})] \leq (1 - p_{int})^2 \cdot m \cdot d$*

This bound can seem quite loose. However from Theorem 3 of Chevalley et al. (2025c), the general bound for a graph is $\mathbb{E}[D_{top}(\mathcal{G}, \pi_{opt})] \leq \sum_{(i,j) \in \mathcal{G}} (1 - p_{int})^{|\mathbf{Pa}_j^{\mathcal{G}} \cup \{j\} \setminus \mathbf{Pa}_i^{\mathcal{G}}|}$. Furthermore, for each edge in a BA graph, we have $|\mathbf{Pa}_j^{\mathcal{G}} \cup \{j\} \setminus \mathbf{Pa}_i^{\mathcal{G}}| \leq 1 + m$, which is close to 2 for a sparse graph. Moreover, for the edge case where $\kappa = 0$, and thus there is a single a single node that is the parent of all the other nodes, the bound is exact. In any case, the bound is $O(d)$.

**Corollary 25.** *The expected FNR is $\mathbb{E}\left[\frac{D_{top}(\mathcal{G}, \pi_{opt})}{|E|}\right] \leq (1 - p_{int})^2$ and the probability bound on the fraction of topological errors is $P\left(\frac{D_{top}(\mathcal{G}, \pi_{opt})}{|E|} \geq c_e\right) \leq \frac{(1 - p_{int})^2}{c_e}$.*

## F PROOFS

*Proof of Lemma 3.* Under the stated assumptions and by the result of Lemma 4 in (Chevalley et al., 2025c), for any edge $(i, j) \in E$ the optimal permutation $\pi_{\text{opt}}$ satisfies

$$\pi_{\text{opt}}(i) < \pi_{\text{opt}}(j)$$

if either $j \in \mathcal{I}$ or there exists some $k \in \mathbf{AN}_j^{\mathcal{G}} \setminus \mathbf{AN}_i^{\mathcal{G}}$ such that $k \in \mathcal{I}$. Hence, when flipping the intervention indicator at node $k$, the only edges whose correct orientation might be affected are those for which the intervention status of $k$ is crucial, and a misorientation of an edge affects $D_{\text{top}}$ by 1; define

$$A_k = \left\{(i, j) \in E : \left(k = j \text{ or } k \in \mathbf{AN}_j^{\mathcal{G}} \setminus \mathbf{AN}_i^{\mathcal{G}}\right)\right\}.$$

By the definition of the Lipschitz constant, we have

$$c_k = |A_k|.$$

We now partition $A_k$ into two disjoint subsets:

$$A_k^{(1)} := \{(i, j) \in E : j = k\} \quad \text{and} \quad A_k^{(2)} := \{(i, j) \in E : k \in \mathbf{AN}_j^{\mathcal{G}} \setminus \mathbf{AN}_i^{\mathcal{G}}\}.$$

1. For any edge $(i, j) \in A_k^{(1)}$, we have $j = k$. Since $i$ must be an ancestor of $k$, the number of edges in $A_k^{(1)}$ is at most the number of ancestors of $k$, so

$$|A_k^{(1)}| \leq |\text{Anc}(k)|.$$

2. For any edge $(i, j) \in A_k^{(2)}$, the condition $k \in \mathbf{AN}_j^{\mathcal{G}}$ implies that $j$ is a descendant of $k$. In the worst-case, each descendant of $k$ contributes at most one edge in $A_k^{(2)}$ (this is a conservative bound given that multiple edges may share the same descendant but we assume a one-to-one correspondence in the worst-case scenario). Thus,

$$|A_k^{(2)}| \leq |\text{Desc}(k)|.$$

Combining these two bounds, we get

$$c_k = |A_k| = |A_k^{(1)}| + |A_k^{(2)}| \leq |\text{Anc}(k)| + |\text{Desc}(k)|.$$

This concludes the proof. □

*Proof of Lemma 4.* Under the relaxed $\epsilon$-interventional faithfulness assumption, by Lemma 5 of (Chevalley et al., 2025c), if for any edge $(i, j) \in E$ the condition

$$j \in \mathcal{I} \quad \text{or} \quad \exists k \in \mathbf{Pa}_j^{\mathcal{G}} \setminus \mathbf{Pa}_i^{\mathcal{G}} \text{ with } k \in \mathcal{I}$$

holds, then $\pi_{\text{opt}}(i) < \pi_{\text{opt}}(j)$. Hence, when flipping the intervention indicator of node $k$, the only edges that might be affected are those whose correct orientation is determined by a direct parental relationship involving $k$. Define

$$B_k := \left\{ (i, j) \in E : k = j \text{ or } k \in \mathbf{Pa}_j^{\mathcal{G}} \setminus \mathbf{Pa}_i^{\mathcal{G}} \right\}.$$

Then the Lipschitz constant is given by

$$c_k = |B_k|.$$

We now consider the two components of $B_k$:

1. For edges $(i, j)$ with $j = k$: here, $i$ must be a direct parent of $k$. Hence,

$$|\{(i, j) \in B_k : j = k\}| \leq \text{in-deg}(k).$$

2. For edges $(i, j)$ with $k \in \mathbf{Pa}_j^{\mathcal{G}} \setminus \mathbf{Pa}_i^{\mathcal{G}}$: in this case, $k$ is a direct parent of $j$. Consequently, each such edge corresponds to a unique edge emerging from node $k$ (i.e., an edge for which $k$ is a parent). Therefore,

$$|\{(i, j) \in B_k : k \in \mathbf{Pa}_j^{\mathcal{G}} \setminus \mathbf{Pa}_i^{\mathcal{G}}\}| \leq \text{out-deg}(k).$$

Combining these, we conclude that

$$c_k = |B_k| \leq \text{in-deg}(k) + \text{out-deg}(k).$$

This completes the proof. □

*Proof of Lemma 5.* Let $\mathcal{G} = (V, E)$ and $\mathcal{G}' = (V, E \triangle \{(i, j)\})$ differ only in the indicator of the edge $(i, j)$, and let

$$\pi = \pi_{\text{opt}}(\mathbf{I}, \mathbf{E}), \qquad \pi' = \pi_{\text{opt}}(\mathbf{I}, \mathbf{E}')$$

denote the *uniquely* tie–broken maximisers (Assumption 2). We distinguish three cases according to the intervention pattern for the endpoints.

—

Case 1: $i \notin \mathcal{I}$

Lemma 16 can only use intervened parents to force an orientation. Since $i$ is *not* intervened, adding or deleting the edge $(i, j)$ never adds a new "witness" parent to any edge $(k, j)$. Hence every orientation rule is unchanged and $f(\mathbf{I}, \mathbf{E}) = f(\mathbf{I}, \mathbf{E}')$.

—

Case 2: $i \in \mathcal{I}$ *and* $j \in \mathcal{I}$

Because the child vertex $j$ itself is intervened, $(k, j)$ is already forced to point $k \to j$ in $\pi$; the extra parent $i$ does not alter that decision. Again $f(\mathbf{I}, \mathbf{E}) = f(\mathbf{I}, \mathbf{E}')$.

—

Case 3 (worst-case): $i \in \mathcal{I}, \ j \notin \mathcal{I}$

Now the sufficient condition of Lemma 16 *can* change. For an edge $(k, j) \in E$ define

$$\delta_{kj} \ = \ \mathbf{1}\Big\{ i \notin \mathbf{Pa}_k^{\mathcal{G}} \Big\}.$$

If $\delta_{kj} = 0$ the set $\mathbf{Pa}_j \setminus \mathbf{Pa}_k$ is the same in $\mathcal{G}$ and $\mathcal{G}'$, so $(k, j)$ keeps its orientation. If $\delta_{kj} = 1$ the new parent $i$ *is* added to that set, so $(k, j)$ may flip direction; this contributes at most $1$ to $\big| f(\mathbf{I}, \mathbf{E}) - f(\mathbf{I}, \mathbf{E}') \big|$.

The total number of such edges is

$$|\mathcal{A}_{ij}| \ = \ \sum_{k \,:\, (k,j) \in E} \delta_{kj} \ \leq \ |\mathbf{Pa}_j^{\mathcal{G}}| \ = \ \deg_{\mathrm{in}}(j),$$

hence $\big| f(\mathbf{I}, \mathbf{E}) - f(\mathbf{I}, \mathbf{E}') \big| \leq \deg_{\mathrm{in}}(j)$.

—

Combining the three cases, the coordinate Lipschitz constant is

$$c_{ij} \ = \ \max_{\text{flip of } E_{ij}} \big| f(\mathbf{I}, \mathbf{E}) - f(\mathbf{I}, \mathbf{E}') \big| \ \leq \ \deg_{\mathrm{in}}(j),$$

and the bound is attained only in the worst-case pattern $i \in \mathcal{I}, \ j \notin \mathcal{I}$. $\qquad\square$

*Proof of Lemma 17.* The proof follows directly from the results in (Chevalley et al., 2025c). Under $\epsilon$-interventional faithfulness and the uniform random choice of $\mathcal{I}$, each edge misorientation probability can be bounded by a term involving $(1 - p_{int})^{|\mathbf{AN}_j^{\mathcal{G}} \cup \{j\} \setminus \mathbf{AN}_i^{\mathcal{G}}|}$. We then apply the Markov inequality to obtain the upper bound. $\qquad\square$

*Proof of Corollary 18.* The result is a direct corollary of Theorem 2 of Chevalley et al. (2025c). $\qquad\square$

*Proof of Theorem 7.* From Lemma 3, the orientation of each edge $(i, j)$ can be guaranteed to be correct if either $j$ is intervened upon, or if there is at least one ancestor of $j$ (not an ancestor of $i$) that is intervened upon. Thus, the intervention status of a single node $k$ can influence only those edges that rely on $k$ to establish correctness.

By definition of $c_k$, these are exactly the edges $(i, j)$ for which $k = j$ or $k \in \mathbf{AN}_j^{\mathcal{G}} \setminus \mathbf{AN}_i^{\mathcal{G}}$. Taking the maximum over all $k$ gives $c = \max_k c_k$. Hence, flipping $I_k$ can change $f(\mathbf{I})$ by at most $c_k \leq c$ and $g(\mathbf{I}) = f(\mathbf{I})/|E|$ by at most $c_k/|E| \leq c/|E|$.

Since $\mathbf{I}$ consists of independent Bernoulli variables and $f$ (resp. $g$) satisfies a bounded-differences condition with constants $(c_k)$ (resp. $(c_k/|E|)$), we can apply McDiarmid's inequality. For $f(\mathbf{I})$, we get:

$$P(|f(\mathbf{I}) - \mathbb{E}[f(\mathbf{I})]| \geq t) \leq 2 \exp\left( -\frac{2t^2}{\sum_{k=1}^{|V|} c_k^2} \right) \leq 2 \exp\left( -\frac{2t^2}{|V| c^2} \right).$$

For $g(\mathbf{I}) = f(\mathbf{I})/|E|$, applying McDiarmid's with differences bounded by $c_k/|E|$, we have:

$$P(|g(\mathbf{I}) - \mathbb{E}[g(\mathbf{I})]| \geq t) \leq 2 \exp\left( -\frac{2t^2}{\sum_{k=1}^{|V|} (c_k/|E|)^2} \right) = 2 \exp\left( -\frac{2|E|^2 t^2}{\sum_{k=1}^{|V|} c_k^2} \right) \leq 2 \exp\left( -\frac{2|E|^2 t^2}{|V| c^2} \right).$$

This completes the proof. □

*Proof of Lemma 19.* **Part 1 (Normalized by $\mathbb{E}[|E|]$):**

By Markov's inequality, for a nonnegative random variable $X$ and $a > 0$, we have:

$$P(X \geq a) \leq \frac{\mathbb{E}[X]}{a}.$$

Apply this with $X = D_{top}$ and $a = c_e \mathbb{E}[|E|]$:

$$P\left(\frac{D_{top}}{\mathbb{E}(|E|)} \geq c_e\right) = P(D_{top} \geq c_e \mathbb{E}[|E|]) \leq \frac{\mathbb{E}[D_{top}]}{c_e \mathbb{E}[|E|]}.$$

From Theorem 4 of Chevalley et al. (2025c), we know:

$$\mathbb{E}[D_{top}] \leq \frac{(1 - p_{int})^2}{p_{int}} d.$$

Also,

$$\mathbb{E}[|E|] = p_e \frac{d(d-1)}{2}.$$

Substitute these into the ratio:

$$\frac{\mathbb{E}[D_{top}]}{c_e \mathbb{E}[|E|]} \leq \frac{\frac{(1-p_{int})^2}{p_{int}} d}{c_e \cdot p_e \cdot \frac{d(d-1)}{2}}.$$

Simplify the fraction:

$$= \frac{(1 - p_{int})^2}{p_{int}} \cdot \frac{2}{c_e p_e d(d-1)} \cdot d.$$

Hence:

$$P\left(\frac{D_{top}}{\mathbb{E}(|E|)} \geq c_e\right) \leq \frac{2(1 - p_{int})^2}{c_e p_{int} p_e (d-1)}.$$

**Part 2 (Normalized by $|E|$):**

Normalizing by the random variable $|E|$ is more challenging since $|E|$ is not constant. Consider any $0 < \delta < 1$ and split the event:

$$P\left(\frac{D_{top}}{|E|} \geq c_e\right) = P(D_{top} \geq c_e |E|) \leq P(D_{top} \geq c_e(1 - \delta)\mathbb{E}[|E|]) + P(|E| < (1 - \delta)\mathbb{E}[|E|]).$$

- For the first term, apply Markov's inequality again:

$$P(D_{top} \geq c_e(1 - \delta)\mathbb{E}[|E|]) \leq \frac{\mathbb{E}[D_{top}]}{c_e(1 - \delta)\mathbb{E}[|E|]}.$$

Using the same bounds as before:

$$\mathbb{E}[D_{top}] \leq \frac{(1 - p_{int})^2}{p_{int}} d, \quad \mathbb{E}[|E|] = p_e \frac{d(d-1)}{2}.$$

Substitute:

$$\frac{\mathbb{E}[D_{top}]}{c_e(1 - \delta)\mathbb{E}[|E|]} \leq \frac{\frac{(1-p_{int})^2}{p_{int}} d}{c_e(1 - \delta)p_e \frac{d(d-1)}{2}} = \frac{2(1 - p_{int})^2}{c_e(1 - \delta)p_{int} p_e (d-1)}.$$

- For the second term, $|E|$ is a Binomial random variable with parameters $N = \frac{d(d-1)}{2}$ and $p_e$. A Chernoff bound gives:

$$P(|E| < (1 - \delta)\mathbb{E}[|E|]) \leq \exp\left(-\frac{\delta^2 \mathbb{E}[|E|]}{2}\right) = \exp\left(-\frac{\delta^2 p_e d(d-1)}{4}\right).$$

Combining both results:

$$P\left(\frac{D_{top}}{|E|} \geq c_e\right) \leq \frac{2(1-p_{int})^2}{c_e(1-\delta)p_{int}p_e(d-1)} + \exp\left(-\frac{\delta^2 p_e d(d-1)}{4}\right).$$

This completes the proof. □

*Proof of Lemma 20.* **Step 1. Concentration for $|E|$.** Let $\mu_d = \mathbb{E}[|E|] = \frac{p_e}{2}d(d-1)$. By the multiplicative Chernoff bound, for any $\delta_d \in (0,1)$,

$$\mathbb{P}(|E| < (1-\delta_d)\mu_d) \leq \exp\left(-\frac{\mu_d \delta_d^2}{2}\right).$$

Set $A_{\delta_d} := \{|E| \geq (1-\delta_d)\mu_d\}$ so that $\mathbb{P}(A_{\delta_d}^c) \leq \exp\left(-\frac{\mu_d \delta_d^2}{2}\right)$.

**Step 2. Bounding the ratio on $A_{\delta_d}$ and on $A_{\delta_d}^c$.** We use the deterministic upper bound $D_{\text{top}} \leq \frac{(1-p_{int})^2}{p_{int}} d$. Thus, on $A_{\delta_d}$,

$$\frac{D_{\text{top}}}{|E|} \leq \frac{\frac{(1-p_{int})^2}{p_{int}}d}{(1-\delta_d)\mu_d} = \frac{(1-p_{int})^2}{p_{int}} \cdot \frac{d}{(1-\delta_d)\frac{p_e}{2}d(d-1)} = \frac{2(1-p_{int})^2}{(1-\delta_d)\,p_e\,p_{int}} \cdot \frac{1}{d-1}.$$

On $A_{\delta_d}^c$, we trivially have $\frac{D_{\text{top}}}{|E|} \leq 1$.

**Step 3. Taking expectations.** Decomposing over $A_{\delta_d}$ and $A_{\delta_d}^c$,

$$\mathbb{E}\left[\frac{D_{\text{top}}}{|E|}\right] \leq \frac{2(1-p_{int})^2}{(1-\delta_d)\,p_e\,p_{int}} \cdot \frac{1}{d-1}\,\mathbb{P}(A_{\delta_d}) + 1 \cdot \mathbb{P}(A_{\delta_d}^c).$$

Since $\mathbb{P}(A_{\delta_d}) \leq 1$ and $\mathbb{P}(A_{\delta_d}^c) \leq \exp\left(-\frac{\mu_d \delta_d^2}{2}\right)$, we obtain

$$\mathbb{E}\left[\frac{D_{\text{top}}}{|E|}\right] \leq \frac{2(1-p_{int})^2}{(1-\delta_d)\,p_e\,p_{int}} \cdot \frac{1}{d-1} + \exp\left(-\frac{\mu_d \delta_d^2}{2}\right),$$

which is the claimed bound.

**Choice of $\delta_d$.** Taking any $\delta_d \downarrow 0$ improves the leading constant toward $\frac{2(1-p_{int})^2}{p_e p_{int}}$, while the tail remains exponentially small provided $\mu_d \delta_d^2 \to \infty$. For example, $\delta_d = d^{-1/2}$ gives $\mu_d \delta_d^2 = \Theta(d)$ and thus an $\exp(-\Omega(d))$ tail, yielding

$$\mathbb{E}\left[\frac{D_{\text{top}}}{|E|}\right] \leq \frac{2(1+o(1))(1-p_{int})^2}{p_e p_{int}} \cdot \frac{1}{d} + \exp\left(-\Omega(d)\right). \qquad \square$$

*Proof of Theorem 8.* **(a)Unnormalised error.** Flipping any single coordinate affects at most all edges incident on one vertex, so $|\Delta f| \leq d$. With $N = \binom{d}{2} + d = \Theta(d^2)$ coordinates, $\sum_\ell c_\ell^2 \leq C_1 d^4$ for a universal constant $C_1$. McDiarmid's inequality yields the stated tail.

**(b) Normalised error.**

Since $0 \leq g \leq 1$, Bhatia–Davis with $(m, M) = (0, 1)$ gives

$$\text{Var}(g) \leq (M - \mathbb{E}g)(\mathbb{E}g - m) = (1 - \mathbb{E}g)\,\mathbb{E}g \leq \mathbb{E}g.$$

Chebyshev's inequality then implies

$$\mathbb{P}\big(|g - \mathbb{E}g| \geq t\big) \leq \frac{\text{Var}(g)}{t^2} \leq \frac{\mathbb{E}g}{t^2}.$$

Finally, insert the expectation bound (Lemma 20),

$$\mathbb{E}g \leq \frac{2(1-p_{int})^2}{(1-\delta_d)\,p_e\,p_{int}} \cdot \frac{1}{d-1} + \exp\left(-\frac{\mu_d \delta_d^2}{2}\right).$$

Choosing $\delta_d = d^{-1/2}$ makes the exponential term $\exp(-\Omega(d))$, and the leading term decays like $1/d$, as stated.

□

*Proof of Lemma 21.* The proof of both bounds follows from Lemma 19 by replacing $p_e$ by $\frac{c}{d}$. For the limits at infinity, we use Lemma 6 of Chevalley et al. (2025c), which states that:

$$\lim_{d\to\infty} \frac{\mathbb{E}[D_{top}(\mathcal{G}, \pi_{opt})]}{d} \leq \frac{(1-p_{int})^2}{p_{int}} \left[1 - \frac{1}{p_{int}\cdot c}\left(1 - e^{-p_{int}\cdot c}\right)\right].$$

The bound at infinity for the first part, i.e. normalized by the expected number of edges, follows directly by applying markov and using Lemma 6. For the second bound, where we normalized by the actual number of edges, we do the same, with the extra step of taking $\delta(d) = d^\alpha$, where $-\frac{1}{2} < \alpha < 0$. Then taking the Taylor expansion of $\frac{1}{1-\delta(d)}$ around 0, only the first term, which is the same as in the first part, does not vanish to 0. $\qquad\square$

*Proof of Lemma 22.* **Step 1. Lower-tail concentration for $|E|$.** By multiplicative Chernoff, for any $\delta_d \in (0, 1)$,

$$\mathbb{P}(|E| < (1-\delta_d)\mu_d) \leq \exp\left(-\frac{\mu_d\delta_d^2}{2}\right), \qquad \mu_d = \frac{c}{2}(d-1).$$

Let $A_{\delta_d} = \{|E| \geq (1-\delta_d)\mu_d\}$, so $\mathbb{P}(A_{\delta_d}^c) \leq \exp(-\mu_d\delta_d^2/2)$.

**Step 2. Bounding the ratio on $A_{\delta_d}$ and $A_{\delta_d}^c$.** We use the deterministic bound $D_{\text{top}} \leq \frac{(1-p_{int})^2}{p_{int}}d$. On $A_{\delta_d}$,

$$\frac{D_{\text{top}}}{|E|} \leq \frac{\frac{(1-p_{int})^2}{p_{int}}d}{(1-\delta_d)\mu_d} = \frac{(1-p_{int})^2}{p_{int}} \cdot \frac{d}{(1-\delta_d)\frac{c}{2}(d-1)} = \frac{2(1-p_{int})^2}{(1-\delta_d)\,c\,p_{int}} \cdot \frac{d}{d-1}.$$

On $A_{\delta_d}^c$, trivially $\frac{D_{\text{top}}}{|E|} \leq 1$.

**Step 3. Taking expectations.** Decomposing over $A_{\delta_d}$ and $A_{\delta_d}^c$ gives

$$\mathbb{E}\left[\frac{D_{\text{top}}}{|E|}\right] \leq \frac{2(1-p_{int})^2}{(1-\delta_d)\,c\,p_{int}} \cdot \frac{d}{d-1}\,\mathbb{P}(A_{\delta_d}) + \mathbb{P}(A_{\delta_d}^c),$$

and the claim follows since $\mathbb{P}(A_{\delta_d}) \leq 1$ and $\mathbb{P}(A_{\delta_d}^c) \leq \exp(-\mu_d\delta_d^2/2)$. $\qquad\square$

*Proof of Corollary 23.* Pick any $\delta_d \downarrow 0$ with $\mu_d\delta_d^2 \to \infty$ (e.g. $\delta_d = d^{-1/2}$). By Lemma 22,

$$\mathbb{E}\big[g(\mathbf{I}, \mathbf{E})\big] \leq \frac{2(1-p_{int})^2}{(1-\delta_d)c\,p_{int}} \cdot \frac{d}{d-1} + \exp\big(-\mu_d\delta_d^2/2\big).$$

Letting $d \to \infty$ sends the exponential term to 0, $\frac{d}{d-1} \to 1$, and $\frac{1}{1-\delta_d} \to 1$. Finally, plug in the scale-free limit for the per-node misorientations (Lemma 6 of Chevalley et al. (2025c)), namely

$$\lim_{d\to\infty} \frac{\mathbb{E}[D_{\text{top}}(\mathcal{G}, \pi_{\text{opt}})]}{d} \leq \frac{(1-p_{int})^2}{p_{int}} \left[1 - \frac{1}{p_{int}c}\left(1 - e^{-p_{int}c}\right)\right],$$

and combine with $\mathbb{E}[|E|] \sim \mu_d = \frac{c}{2}(d-1)$ as in the proof of Lemma 22 to obtain the stated bound. $\qquad\square$

*Proof of Theorem 9.* We apply Warnke's 0–1 *typical bounded differences* inequality (Theorem 13) to the families $\{E_{ij}\}_{i<j}$ and $\{I_k\}_{k=1}^d$ jointly.

**1. High–probability event $\Gamma$.** Define

$$\Gamma := \Big\{\deg_{\max} \leq K_0 \log d\Big\} \cap \Big\{|E| \leq 2cd\Big\},$$

where $K_0 > 0$ is chosen so that $P(\neg\Gamma) \leq d^{-(K+4)}$, $K > 0$. The two parts of $\Gamma$ follow from Chernoff bounds and a union bound over vertices.

**2. Typical and worst–case Lipschitz constants.**

$$\text{edge } (i,j): \quad c_{ij} = K_0 \log d \, \mathbf{1}_\Gamma, \qquad d_{ij} = d;$$
$$\text{intervention } I_k: \quad c_k = K_0 \log d \, \mathbf{1}_\Gamma, \qquad d_k = d.$$

Indeed, on $\Gamma$ flipping *any* coordinate changes $f$ by at most $\deg_{\max} \leq K_0 \log d$, whereas in the worst case it may change as many as $d$ edges.

**3. Parameters $\gamma_k$, compensations $e_k$, and $y_k$.** Choose the *uniform* parameter

$$\gamma_k := d^{-2} \quad (\forall k), \qquad e_k := \gamma_k(d_k - c_k) = \frac{d - K_0 \log d \, \mathbf{1}_\Gamma}{d^2}.$$

**4. Bad event $\mathcal{B}$.** Warnke's bound gives

$$P(\mathcal{B}) \leq \sum_{k=1}^{N} \frac{P(\neg\Gamma)}{\gamma_k} \leq \left(\Theta(d^2)\right) d^2 \, d^{-(K+4)} \leq d^{-K}.$$

**5. Constants $C$ and $V$ for the two functionals.** *(i) Unnormalised error $f$.*

- $C := \max_k(c_k + e_k) \leq K_0 \log d + (d - K_0 \log d)/d^2 \leq c_2 \log d$ for some $c_2 > 0$.

- Edge variables: $p_e = c/d$, one–bit variance $(1 - p_e)p_e(c_k + e_k)^2 = O\!\left(\frac{c}{d}(\log d)^2\right)$. With $\binom{d}{2}$ edges this sums to $\Theta(d \log^2 d)$.

- Intervention variables: $p_{\text{int}}$ constant, $d$ terms of order $(\log d)^2$, yielding the same order. Hence $V = \Theta(d \log^2 d)$.

Warnke's inequality gives the tail in part (a).

*(ii) Normalised error $g = f/|E|$.*

On $\Gamma$, $|E| \leq 2cd$, dividing the $c_k$ and $d_k = 1$. $C$ and every summand of $V$ by $|E|^{-2} = O(d^{-2})$:

- $C = O\!\left(\frac{\log d}{d}\right) = c_4 \frac{\log d}{d}$.

- Edge contribution to $V$: $O\!\left(\frac{c}{d} \cdot \frac{\log^2 d}{d^2}\right)$ times $\binom{d}{2}$, i.e. $O\!\left(\frac{\log^2 d}{d}\right)$.

- Intervention contribution: $d$ terms, each $O\!\left(\frac{\log^2 d}{d^2}\right)$, same order.

Thus $V = O\!\left(\frac{\log^2 d}{d}\right)$ and Warnke's 0–1 inequality yields the bound in part (b).

$\square$

*Proof of Theorem 11.* **Step 1: High-Probability Degree Bound.** By construction in the generalized Barabási–Albert model, every node (beyond the initial seed) has in-degree exactly $m$. Existing results show that with high probability the out-degree of every node is at most $C \, d^\beta$, where

$$\beta = \frac{1}{\gamma - 1} \quad \text{and} \quad \gamma = 2 + \frac{\kappa}{m}.$$

That is, with high probability, the entire graph satisfies

$$\text{in-deg}(i) + \text{out-deg}(i) \leq m + C \, d^\beta, \qquad \forall i \in V.$$

We denote this event by $\mathcal{E}_{BA}$ and its support by $E_{BA}$.

**Step 2: Bounded Differences (Conditional on $\mathbf{E} \in E_{BA}$).** Fix any graph $\mathbf{E} \in E_{BA}$. Then the graph is deterministic, and only the intervention vector $\mathbf{I} = (I_1, \ldots, I_d)$ is random, with each $I_i$ being an independent Bernoulli($p_{int}$) variable. Define

$$f(\mathbf{I}) \; := \; D_{\text{top}}\big(\mathcal{G}, \pi_{\text{opt}}(\mathbf{I})\big),$$

where $\mathcal{G}$ is the DAG determined by $\mathbf{E}$. For each node $i$, let

$$c_i \; := \; \max_{\substack{\mathbf{I}, \mathbf{I}' \in \{0,1\}^d \\ I_j = I_j' \text{ for all } j \neq i}} \Big| f(\mathbf{I}) - f(\mathbf{I}') \Big|.$$

Under the "parents-only local influence" assumption—that is, flipping $I_i$ can only affect the orientations of edges incident on node $i$—we have for any fixed $\mathbf{E} \in E_{BA}$

$$c_i \leq \text{in-deg}(i) + \text{out-deg}(i) \leq m + C\, d^\beta.$$

Thus, for all $i$ we may take

$$c_i \leq m + C\, d^\beta.$$

Consequently, the sum of squares is bounded by

$$\sum_{i=1}^{d} c_i^2 \leq d \left( m + C\, d^\beta \right)^2.$$

**Step 3: Application of McDiarmid's Inequality (Unnormalized Error).** Conditional on the fixed graph $\mathbf{E} \in E_{BA}$, the intervention vector $\mathbf{I}$ consists of independent random variables. Hence, by McDiarmid's inequality,

$$P\Big( \big| f(\mathbf{I}) - \mathbb{E}[f(\mathbf{I}) \mid \mathbf{E}] \big| \geq t \;\Big|\; \mathbf{E} \Big) \; \leq \; 2 \exp\!\Big( -\frac{2t^2}{\sum_{i=1}^{d} c_i^2} \Big) \; \leq \; 2 \exp\!\Big( -\frac{2t^2}{d \left( m + C\, d^\beta \right)^2} \Big).$$

Thus, for every fixed graph $\mathbf{E} \in E_{BA}$, $f(\mathbf{I})$ concentrates around its conditional mean $\mathbb{E}[f(\mathbf{I}) \mid \mathbf{E}]$ with deviation scale $O\big( d^{\beta + \frac{1}{2}} \big)$.

**Step 4: Application for the Normalized Error.** Define the normalized topological error by

$$g(\mathbf{I}) \; := \; \frac{f(\mathbf{I})}{|E|} \; = \; \frac{f(\mathbf{I})}{m\, d},$$

since by construction, $|E| = m\, d$ (when starting from an empty seed). For any fixed $\mathbf{E} \in E_{BA}$, flipping the intervention $I_i$ changes $f(\mathbf{I})$ by at most $m + C\, d^\beta$; hence, the corresponding change in $g(\mathbf{I})$ is at most

$$\frac{m + C\, d^\beta}{m\, d}.$$

Since this bound holds for each $i$, the sum of squares of these differences is

$$\sum_{i=1}^{d} \left( \frac{m + C\, d^\beta}{m\, d} \right)^2 = d \left( \frac{m + C\, d^\beta}{m\, d} \right)^2 = O\big( d^{2\beta - 1} \big).$$

Thus, applying McDiarmid's inequality conditionally on $\mathbf{E}$ yields, for any $t > 0$,

$$P\Big( \big| g(\mathbf{I}) - \mathbb{E}[g(\mathbf{I}) \mid \mathbf{E}] \big| \geq t \;\Big|\; \mathbf{E} \Big) \; \leq \; 2 \exp\!\Big( -\frac{2t^2}{O(d^{2\beta - 1})} \Big).$$

That is, $g(\mathbf{I})$ concentrates around its conditional mean $\mathbb{E}[g(\mathbf{I}) \mid \mathbf{E}]$ at the scale $O\big( d^{\beta - \frac{1}{2}} \big)$. Notably, if $\gamma = 3$ (so that $\beta = \frac{1}{2}$), the concentration is at an $O(1)$ level; if $\gamma > 3$ (so that $\beta < \frac{1}{2}$), the normalized error becomes even tighter.

**Conclusion.** The proof shows that for any fixed graph $\mathbf{E} \in E_{BA}$ (i.e., in the high-probability event $\mathcal{E}_{BA}$), the deviation of the topological error $f(\mathbf{I})$ (and hence $g(\mathbf{I})$) from its conditional mean is bounded by the stated exponential terms. The overall statement then holds with high probability / for almost every graph (over the random graph generation) for every such fixed graph.

This completes the proof. $\qquad\qquad\square$

# G    ADDITIONAL EMPIRICAL RESULTS

We provide supplementary plots corresponding to the parameter sweep described in Section 5.

## G.1    DEVIATION PLOTS USING THE STANDARD DEVIATION OF FNR

Figure 2 replicates the analysis of Figure 1, but reports the standard deviation instead of the interquartile range (IQR). The qualitative behavior is the same: variability decreases with $d$ for all graph families, consistent with our theoretical predictions.

## G.2    EMPIRICAL EVALUATION OF THE SCALING OF THE MEAN ERROR OF FNR

Using the same parameter sweep as in Section 5, Figure 3 plots the mean false negative rate (FNR) as a function of graph size $d$, alongside the theoretical upper bounds derived in Appendix E. Across all graph families and parameter settings, the empirical means remain below the theoretical bounds, confirming their validity. The only visible exception occurs at high intervention coverage ($p_{\text{int}} = 0.75$), where the empirical error occasionally exceeds the bound. One possible explanation is that in this regime the distance matrix becomes denser, which increases the difficulty of the optimization and may cause DiffIntersort to return orderings farther from the true optimum. This highlights that our results analyze *theoretical properties of the score function*, while the empirical performance also depends on the optimization dynamics of the chosen algorithm.

## G.3    ADDITIONAL RESULTS FOR THE UNNORMALIZED ERROR $D_{\text{TOP}}$

For completeness, we report the same set of empirical results as in Section 5, but using the unnormalized topological error $D_{\text{top}}(\mathcal{G}, \pi_{\text{opt}})$ instead of the normalized false negative rate (FNR). Figures 4 to 6 show the mean values and deviation widths of $D_{\text{top}}$ across the three graph families (ER, scale-free ER, BA) under the same parameter sweep.

## G.4    SCALING OF MAXIMUM DEGREE IN BA GRAPHS

Figure 7 examines how the empirical scaling of the maximum degree in Barabási–Albert (BA) graphs approaches the theoretical prediction. We generate graphs with $m = 3$ edges per node and consider $\kappa \in \{1.0, 3.0, 9.0\}$. Theory predicts degree exponents $\gamma = \frac{7}{3}, 3, 5$, while the empirical fits yield $\hat{\gamma} = 2.37, 2.93, 4.11$, respectively. The estimates align closely with the theoretical values, with only a mild underestimation for $\kappa = 9.0$, suggesting slower convergence in this regime. Overall, these results validate that our generated BA graphs faithfully reproduce the expected heavy-tailed behavior.

# H    ANALYSIS ON SYNTREN-STYLE NETWORKS

To complement our analysis on parametric random-graph ensembles, we perform an additional study on biologically inspired synthetic networks generated using the SynTReN model (Van den Bulcke et al., 2006). These networks do not follow a closed-form generative model (e.g. ER or generalized BA), and therefore our parametric deviation theorems cannot be applied to them. However, the general fixed-graph deviation inequality (Theorem 7) together with the expectation upper bound of Corollary 18 hold for *any* fixed DAG. We also apply the parent-only local influence assumption. This makes it possible to assess whether the dimension-adaptive behavior observed on ER/BA persists on more realistic network topologies.

**Datasets.**    We extract gene-regulatory networks of sizes $d = 300$ and $d = 1000$ from the `grndata` Bioconductor package, which provides SynTReN-style topologies.* We construct the corresponding DAGs, removing edges when necessary to avoid cycles. The resulting graphs, which we refer to as SYNTREN300 and SYNTREN1000, contain realistic regulatory motifs but do not arise from a known parametric distribution.

---

*`https://bioconductor.org/packages/grndata`

**Results.** Across all metrics—expectation bounds, deviation bounds, and large-error probability bounds—SYNTREN1000 exhibits consistently *smaller* orientation error than SYNTREN300 for all values of $p_{int}$ (Figure 8). Specifically:

- the upper bound on $\mathbb{E}[g(\mathbf{I})]$ is uniformly lower in the $d = 1000$ network;

- the deviation width shrinks as the network dimension increases;

- consequently, the bound on $P(g(\mathbf{I}) \geq 0.4)$ is much smaller for SYNTREN1000 across all examined $p_{\text{int}}$.

These findings align with the dimension-adaptive behavior predicted by our parametric results: larger networks exhibit both lower or equal expected orientation error and tighter concentration, even when the graph does not follow a simple random-graph distribution. This provides empirical support that the stabilizing effect of dimension, established analytically for ER and BA distributions, also appears in realistic biological network topologies under randomized interventions.

## I  LLM USAGE DECLARATION

We declare that LLM systems were used to perform part of this work, such as polishing the text and presentation, writing (plotting) scripts, and exploring potential proof techniques.

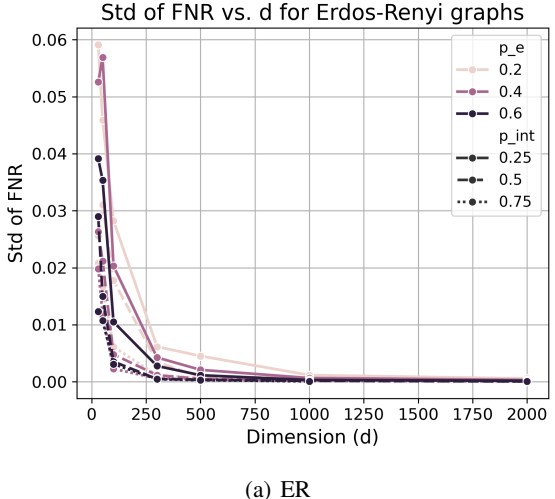

(a) ER

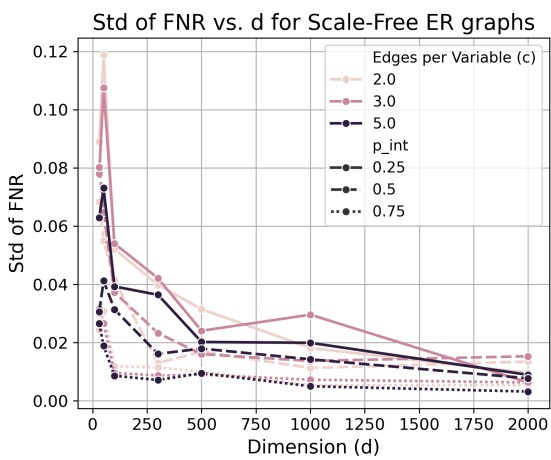

(b) Scale-free ER

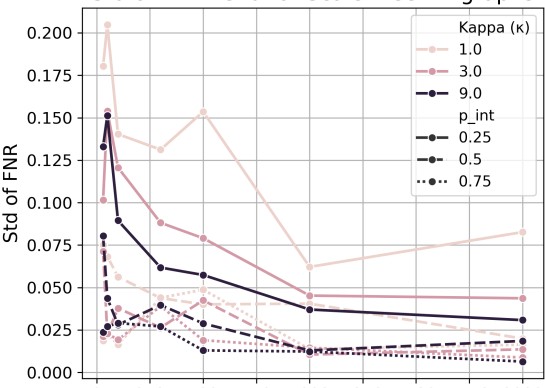

(c) Scale-free BA

Figure 2: Standard deviation of the FNR as a function of graph size $d$. For each graph family, results are shown across three density parameters and three values of intervention coverage $p_{\text{int}}$. The deviation vanishes with growing $d$, in line with theory, except for scale-free BA graphs with $\kappa = 1$, corresponding to a heavy-tailed regime with exponent $\gamma = \frac{7}{3} < 3$.

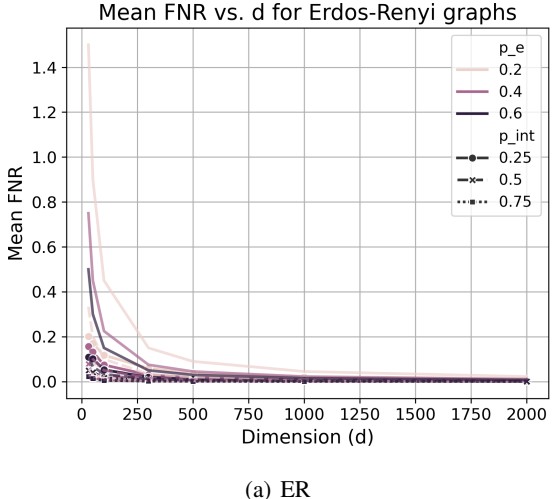

(a) ER

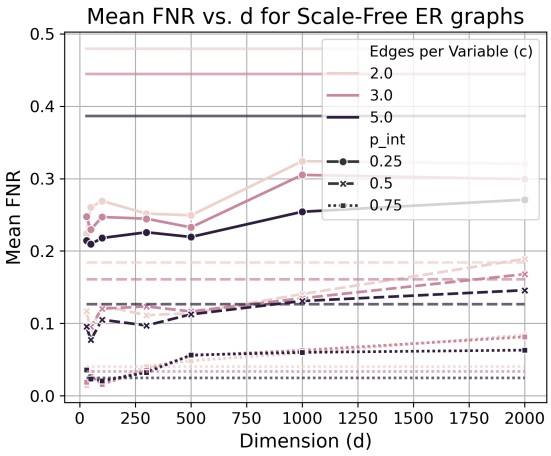

(b) Scale-free ER

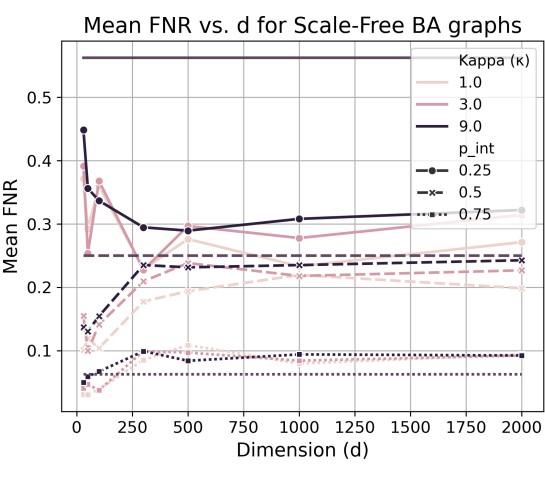

(c) Scale-free BA

Figure 3: Mean false negative rate (FNR) versus graph size $d$ across Erdős–Rényi (ER), scale-free ER, and Barabási–Albert (BA) graphs. The solid lines with points denote empirical averages; lines without points show theoretical upper bounds from Appendix E. The bounds hold across all settings, with a slight mismatch at high intervention coverage ($p_{\mathrm{int}} = 0.75$), likely due to optimization difficulties in DiffIntersort.

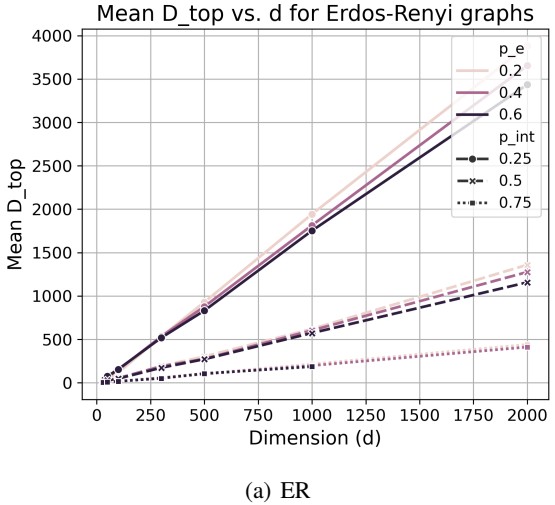

(a) ER

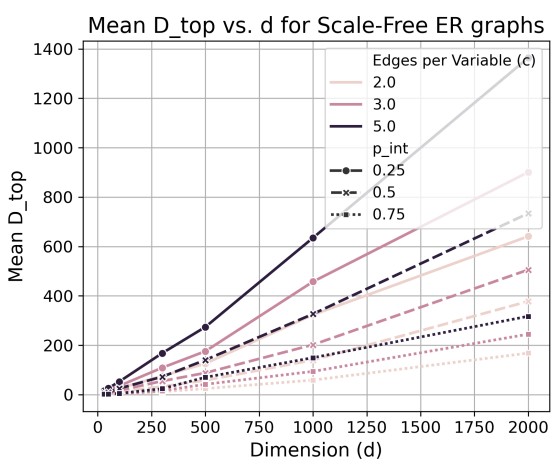

(b) Scale-free ER

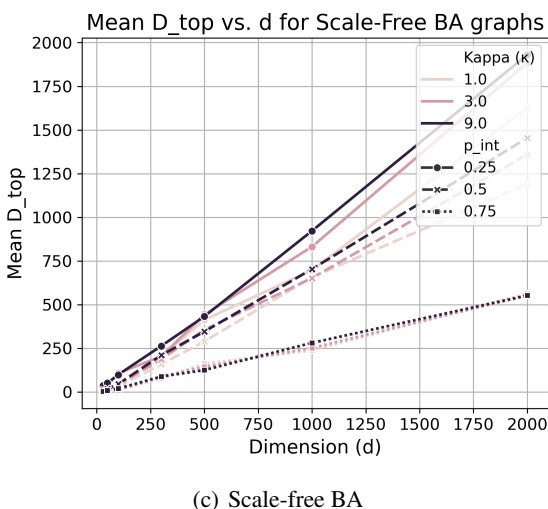

(c) Scale-free BA

Figure 4: Mean unnormalized error $D_{\text{top}}$ versus graph size $d$ across Erdős–Rényi (ER), scale-free ER, and Barabási–Albert (BA) graphs.

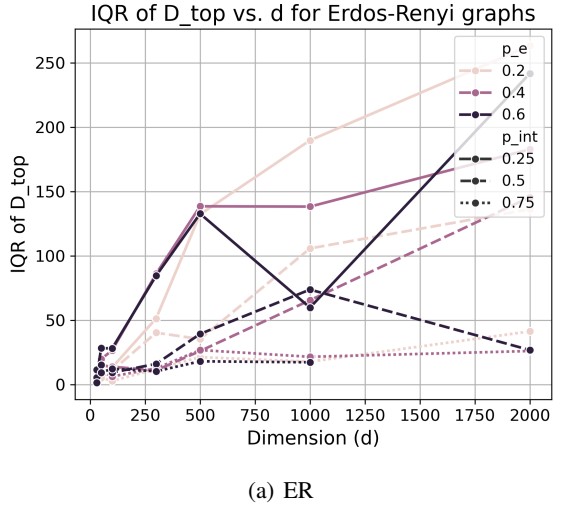

(a) ER

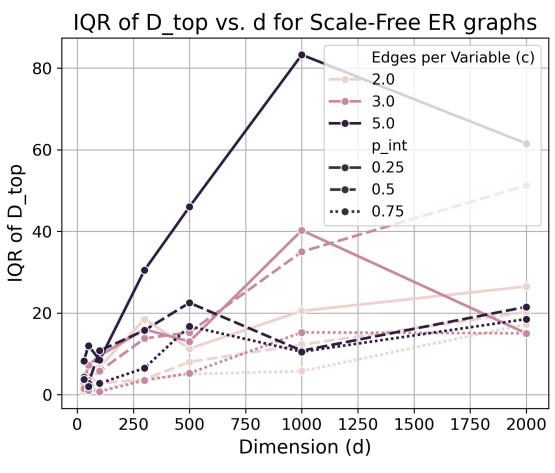

(b) Scale-free ER

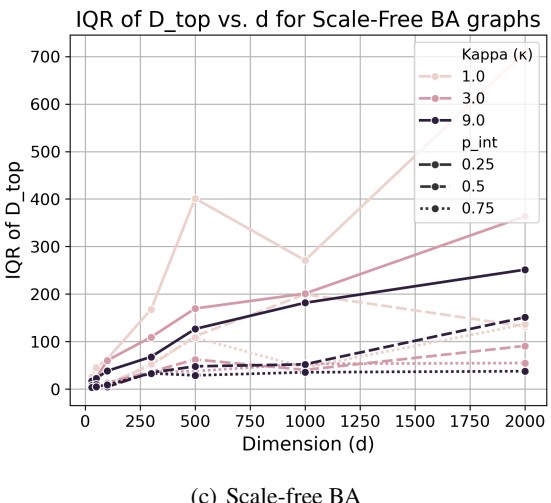

(c) Scale-free BA

Figure 5: IQR of unnormalized error $D_{\text{top}}$ versus graph size $d$ across Erdős–Rényi (ER), scale-free ER, and Barabási–Albert (BA) graphs.

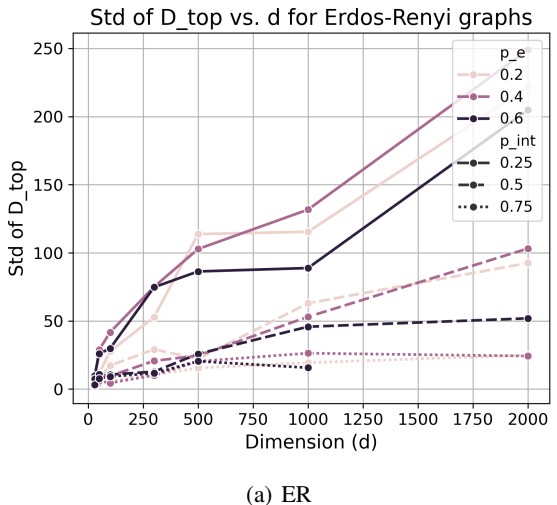

(a) ER

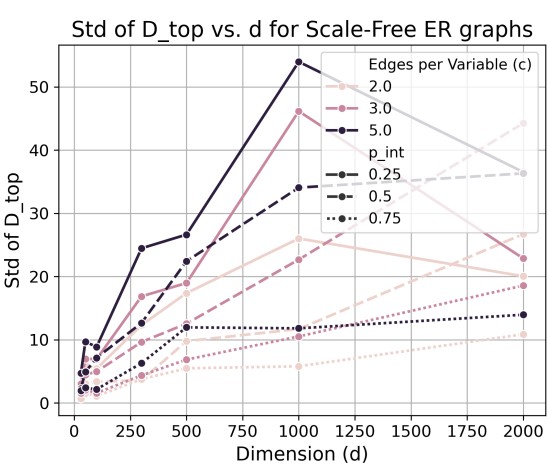

(b) Scale-free ER

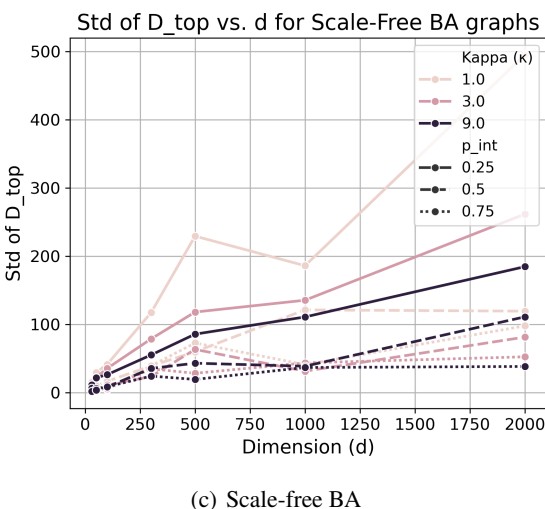

(c) Scale-free BA

Figure 6: Standard deviation of unnormalized error $D_{\text{top}}$ versus graph size $d$ across Erdős–Rényi (ER), scale-free ER, and Barabási–Albert (BA) graphs.

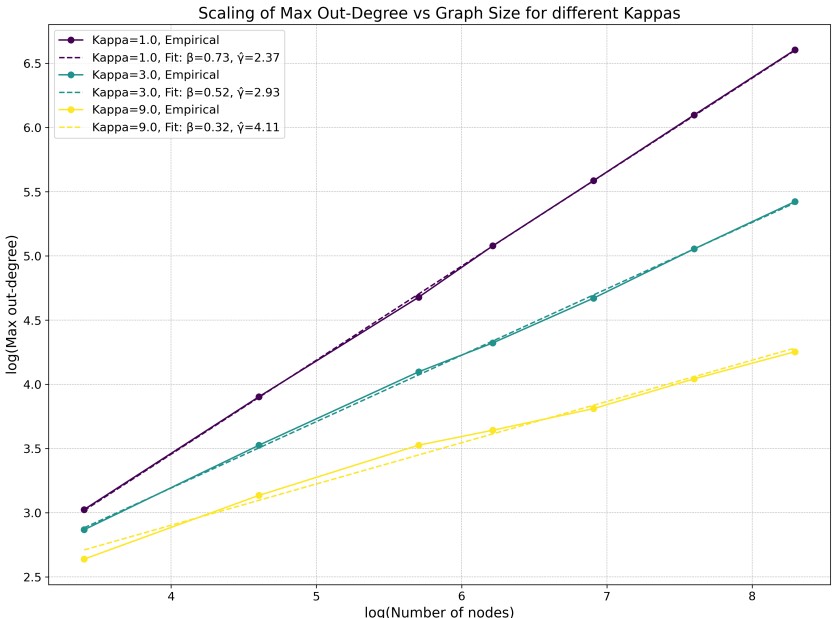

Figure 7: Empirical scaling of the maximum degree in BA graphs with $m = 3$ edges per node for different values of $\kappa$. Fitted lines correspond to estimated exponents $\hat{\gamma}$, compared against theoretical predictions. Graph sizes range from $d = 30$ to $d = 4000$. The close match confirms that the generated graphs reproduce the expected heavy-tailed scaling.

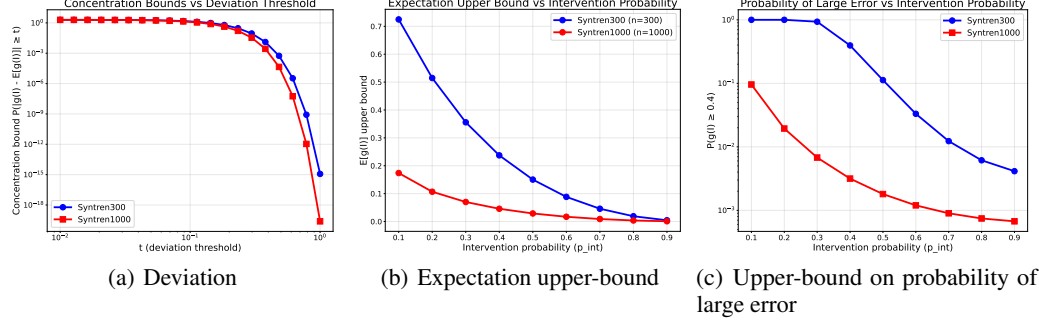

(a) Deviation      (b) Expectation upper-bound      (c) Upper-bound on probability of large error

Figure 8: Comparison of the upper-bounds on the deviation, expectation and probability of large errors between SYNTREN300 and SYNTREN1000 networks (Van den Bulcke et al., 2006). For the deviation plot (a), we have $t \in [0.1, 1.0]$ for 20 values on a logarithmic scale. For (b) and (c), we have $p_{int} \in [0.1, 0.2, 0.3, 0.4, 0.5, 0.6, 0.7, 0.8, 0.9]$.

