# OpenReview forum: "Theoretical Guarantees for Causal Discovery on Large Random Graphs"
_ICLR.cc/2026/Conference — ICLR 2026 Poster_

### Official Review · Reviewer_ifw6 · 2025-10-28

**Soundness:** 3
**Presentation:** 2
**Contribution:** 2
**Rating:** 4
**Confidence:** 3

**Summary:**

The authors derive high-probability deviation bounds for the (normalized) number of true edges missed due to incorrect causal ordering — that is, for the false negative rate — in the context of causal discovery with interventional data on random graphs. Their theory applies to the score-optimal causal order (as defined by the score function introduced in Chevalley et al., 2025c) and covers both Erdős–Rényi and Barabási–Albert graph models.

**Strengths:**

The presentation of the technical contributions of the paper is clear. The technical contribution themselves seem solid (although I haven’t checked the proofs myself). The empirical evidence of figure 1 convingly support the theory.

**Weaknesses:**

- **Presentation.**
    - In the abstract and introduction, it is not clear that the FNR concentration refers to the causal order that optimizes the specific score in Eq. (2) (this only becomes precise later via the notation and Assumption 2). This oversells the scope: the bounds apply to the score-optimal order for Eq. (2).
    - The goal is scattered: L48–49 suggests a broad “analysis of interventional CD,” whereas L52–53 focuses on finite-sample deviation bounds for FNR. Please state *up front* (abstract + intro) that the paper derives deviation bounds for $D_{\text{top}}$ and its normalization (FNR) **for the Eq. (2) optimizer**, on ER and BA random graphs. The problem definition currently appears only in Related Work (L135–143); it belongs in the introduction.
- **Motivation**. I’d appreciate a clearer case for the value and motivation of this work.
    - Why does the authors believe that it is valuable to prove this results for optimizers of the score of eq. (2)? Should we expect similar behaviour for the causal order found by other interventional causal discovery methods? Can we show that empirically?
    - Why do the authors believe that it is interesting to study concentration bounds for synthetically generated graphs? Would the theoretical results on ER/BA be relevant for real world causal discovery? Can we have an empirical analysis on real world / semi synthetic data (e.g. syntren) supporting the claim that these results would be useful for real world problems?

    The technical contributions are sound. I don’t see why they are valuable for the causal discovery community, due to their narrow scope (they refer specifically to optimizers o the score of eq.(2)) and empirical evidence limited to artificial settings.

**Questions:**

- **Experiments.** I’d like to see the plots of figure 1  also for $D_{top}$, to visualize whether there is any additional information we can gain from knowing concentration results on FNR on top of what we already know for $D_{top}$
- **Assumption 2.** Is there any dependency of the results on the tie breaking rule chosen?
- **L427-428**: authors write that “prior work shows” that DiffiIntersort “achieves close approximations [..] and is scalable”. What prior work?

---

> ### Author Response · Authors · 2025-11-20
>
> We thank the reviewer for the constructive comments and for recognizing the soundness of the technical contributions. We address the raised points below.
>
> (1) Scope and positioning of the results in the abstract and introduction
>
> We believe the abstract and introduction already state the mathematical scope accurately: the results hold under $\epsilon$-interventional faithfulness and random single-variable interventions. Under these assumptions, the false-negative rate is an intrinsic property of the underlying interventional distribution, not of any particular algorithm. The score in Eq. (2) is simply a precise formal device that allows us to express and analyze this quantity.
>
> Therefore, the deviation bounds apply to any dataset or method operating under the same assumptions. If an algorithm exploits these assumptions (or equivalent ones), the theory describes an upper-bound on the optimal behavior it can hope to achieve; if not, the bounds quantify how much performance is potentially left on the table.
>
> (2) Motivation for studying this score and this form of deviation guarantee
>
> The interventional score in Eq. (2) is an object capturing how interventions reveal descendant relationships. It provides a clean and assumption-minimal way to characterize orientation error. The deviation bounds therefore inform on fundamental stability properties of causal discovery under interventions, independent of specific optimization procedures.
> This is why the results are not narrow: they describe a property of the underlying causal model and intervention design, not only of the InterSort algorithm itself.
>
> (3) Synthetic ER/BA graph models and relevance for practice
>
> Our focus on ER and generalized BA models is deliberate: they isolate the combinatorial structural effects (sparsity, scale-free, heavy-tailed degrees) that drive orientation difficulty under interventions. These graph distributions are also the standard baselines used throughout the causal-discovery literature, so understanding their deviation behavior directly informs how practitioners interpret synthetic evaluations.
>
> The results therefore speak both to foundational understanding and to the regimes commonly used in practice.
>
> (4) Experiments on real or semi-synthetic data
>
> We appreciate the suggestion to examine whether the theoretical behaviors appear on semi-synthetic or real-world inspired networks. We performed this analysis using the bioconductor R package grndata, which provides networks in the SynTReN style. We extracted networks with $d=300$ and $d=1000$ nodes and applied our general deviation bound (Theorem 7) together with the expectation upper bound from Corollary 18. For each network, we computed:
>
> - the bound on the expected FNR,
> - the deviation bound for multiple values of t,
> - and the probability of a large error, i.e., $P(FNR \geq 0.4)$.
>
> Across all quantities and all values of $p_{int}$, we observed consistent behavior:
>
> - the larger network exhibits a smaller expected FNR,
> - tighter deviations,
> - and, when combining the two, a substantially lower probability of large orientation error.
>
> This matches the trend predicted by our theory: increasing dimension tends to stabilize orientation error under random interventions, even for networks not sampled from a clean parametric family. Importantly, SynTReN networks do not follow a closed-form generative model, so we cannot derive explicit deviation expressions as we do for ER and BA; only the fixed-graph bounds of Theorem 7 apply. Nevertheless, the empirical application of these bounds on SynTReN-style networks confirms the same dimension-adaptive concentration behavior observed in our parametric models.
>
> (5) Responses to specific questions
>
> - Plots for raw $D_{top}$.
>
> These are already included in the appendix (Figures 5 and 6).
>
> - Assumption 2 and tie-breaking.
>
> The assumption is needed for mathematical precision, and is independent of the rule chosen, but is irrelevant in practice: in our experiments we did not enforce any tie-breaking rule.
>
> - L427–428 (“prior work shows that DiffInterSort achieves close approximations and is scalable”).
>
> This refers to Chevalley et al. (2024), where DiffIntersort was introduced and its approximation accuracy and scalability were empirically validated.
>
>
> We appreciate the reviewer’s suggestions regarding clarification. We hope this resolves the questions raised.

---

> > ### Comment · Reviewer_ifw6 · 2025-11-23
> >
> > I thank the authors for the thorough response. Comments in response:
> >
> > - (1). Thank you for the clarification. There was a misunderstanding on my side about this point.
> > - (3) The authors argue that ER and BA algorithms are *realistic.* This is arbitrary. Neither the paper nor the comments in the rebuttal provide strong ground for this claim:
> >     - I still wonder why are these random graph models realistic for causal discovery problems?
> >     - The authors say that it is standard to use BA and ER graphs for synthetic benchmarking. I’m aware of it, but this is not evidence that BA and ER graphs are realistic for causality.

---

> > > ### Author Response · Authors · 2025-11-25
> > >
> > > A central motivation for using ER- and BA-type ensembles is precisely the one underlying the entire field of random graph theory: real networks are single, heterogeneous realizations of complex and largely unknown generative processes, and therefore cannot meaningfully be modeled by a single “true” distribution. For this reason, network science has long relied on stylized random-graph families to study which structural regimes (such as sparsity, degree heterogeneity, hub formation, or local branching patterns) influence observable behaviors across large classes of real networks. These models are not intended to be literal generative descriptions; they are analytically tractable representatives of recurring structural patterns. ER and generalized BA graphs bracket two fundamental extremes of network structure (homogeneous finite-variance degrees versus heavy-tailed hub-dominated topologies), and the qualitative properties they isolate are precisely those that determine the difficulty of orienting edges under interventions. This is why random graph models are the standard theoretical tool for understanding network phenomena in fields ranging from social and technological networks to biological and neural systems. Our results follow this well-established approach: the goal is not to assert that ER or BA generate real causal graphs, but to clarify how key structural features influence the stability of causal orientation. As you suggested, we also supplemented the analysis with SynTReN-style biological networks, where the same dimension-adaptive behavior appears.
> > >
> > > We revised the manuscript to clarify that ER/BA are used as stylized structural regimes rather than literal generative models, adapting the corresponding phrasing in the abstract, introduction, related work, and conclusion.

---

> > > > ### Comment · Reviewer_ifw6 · 2025-11-26
> > > >
> > > > Thank you to the authors for their response. I follow up with some comments and questions:
> > > >
> > > > > This is why random graph models are the standard theoretical tool for understanding network phenomena in fields ranging from social and technological networks to biological and neural systems
> > > > >
> > > >
> > > > Talking about networks, in my opinion, is too general. What you propose to study are causal graphs, and you do that with a specific task in mind, that is, causal discovery. Having this in mind is what motivates my questions:
> > > >
> > > > - In causal graphs, the meaning of the arrows is given by (i) Markovianity and faithfulness assumptions that connect the factorization of the distribution to that of the graph, plus (ii) some definition of interventional probability. Why are BA/ER random networks a good model to study graphs whose arrows have this meaning? In this regard, the authors write that
> > > >
> > > >     > the qualitative properties they isolate are precisely those that determine the difficulty of orienting edges under interventions.
> > > >     >
> > > >
> > > >     Why is this the case? E.g., say I have the graphs $X_1 \to X_2$ and a graph $X_1 \to X_2 \to X_3$ and a fully connected graph $X_3 \to X_1 \to X_2 \leftarrow X_3$. Why difference in the structure (here, node degrees) make it harder to orient edges under interventions? If this is not a meaningful example, please provide one that clarifies this point convincingly
> > > >
> > > > - Regarding the specific goal of causal discovery, the task can be rephrased as finding the causal direction and a valid adjustment set when I want to study the effect of interventions. With this goal in mind, why is it useful to know that for some classes of random graphs, on average, I can expect a certain error in the orientations, as a function of the graph dimension?  In this case, again, I fail to see the connection between the proposed study on the concentration of errors in high dimensions for these random graphs and the goals of causal discovery that motivate the study.

---

> > > > > ### Author Response · Authors · 2025-11-27
> > > > >
> > > > > (1) On whether biological or neural networks are “causal graphs.”
> > > > >
> > > > > In the structural causal model (SCM) framework, many empirical systems used as motivating examples are causal graphs: their edges correspond to physical mechanisms by which interventions propagate. In gene-regulatory networks, for example, knocking out gene $i$ alters the expression of its regulatory targets through biochemical mechanisms; this is exactly the notion of “$i$ causes $j$” represented in an SCM.  Thus these systems correspond to mechanistic causal structures.
> > > > >
> > > > > (2) Why structural properties affect the difficulty of orienting edges.
> > > > >
> > > > > Once causal relations are represented as a directed graph, the statistical difficulty of orientation from interventions depends on structural quantities such as degrees, descendant-set sizes, and path geometry. For example, if a node $h$ is a hub with many children, then intervening on $h$ produces a distinctive, high-signal cascade of changes across its descendants, immediately orienting many edges at once. In contrast, in low-degree regions or long chains, interventions affect small and overlapping sets of nodes, making neighboring directions harder to distinguish. Therefore, the degree distribution and global topology directly influence how informative each intervention is, and how much the orientation error can fluctuate.
> > > > >
> > > > > (3) Why ER/BA are appropriate for studying these structural effects.
> > > > >
> > > > > Given the mechanistic interpretation above, the relevant question is whether they approximate the topological properties of the underlying causal graph that an SCM assigns to that system. Empirical causal graphs in domains such as gene regulation and neural dynamics are well known to be sparse, modular, and often heavy-tailed, with a mix of low-degree nodes and influential hub nodes. In this sense, ER and generalized BA models serve as good approximations of the topology of many large causal graphs, even though they do not encode the system’s specific causal mechanisms. These stylized ensembles provide analytically tractable families for understanding how such topological features influence the informativeness of randomized interventions and, consequently, the difficulty of orienting causal edges.
> > > > >
> > > > > (4) Why this is useful for causal discovery.
> > > > >
> > > > > We also clarify that causal discovery encompasses more than identifying a causal direction together with an adjustment set. In many scientific domains, recovering the causal structure itself is the primary goal: for example, to understand the mechanisms governing a biological, neural, or engineered system, to generate hypotheses about regulatory or signaling pathways, or to identify potential intervention targets. In these settings, evaluating the quality of the recovered structure is essential, and metrics such as SHD, SID, and the false-negative rate can be used to quantify structural accuracy. Our deviation results therefore analyse the stability of causal structure recovery under randomized interventions.

---

### Official Review · Reviewer_A2f8 · 2025-10-28

[review text omitted: it was posted to a different submission]

---

> ### Author Response · Authors · 2025-11-20
>
> We thank the reviewer for the extensive feedback. Several comments appear to rely on assumptions about the paper’s content that do not apply to our submission, so we provide clarifications below.
>
> (1) The paper does not propose a new causal-discovery algorithm
>
> The submission does not introduce:
> - a new permutation-based DAG parameterization,
> - a differentiable objective for causal discovery,
> - a continuous relaxation analogous to NOTEARS/DAG-GNN/GOLEM,
> - or a new optimization/search procedure.
>
> The paper’s contribution is exclusively theoretical: we derive deviation bounds on the false-negative rate of causal orientation under random single-variable interventions for several random-graph families. The analysis concerns the optimal solution of the InterSort score, not a practical search algorithm. InterSort and DiffIntersort are pre-existing methods used only as references to define the scoring objective and to illustrate trends empirically.
>
> Consequently, issues such as local minima, search behavior, scoring design, and algorithmic comparisons are not in the scope of this paper.
>
> (2) Claims of scalability
>
> The review states that empirical results do not extend beyond “a few hundred variables,” but all figures run DiffIntersort up to 2000 variables, and this is stated in the text.
>
> (3) Connection between theory and optimization
>
> The review requests clarification on how guarantees apply to the search procedure. Again, we do not analyze an optimization method. The guarantees concern the score-maximizing causal order, following the standard approach in identifiability analyses. Properties of gradient-based optimization algorithms are outside the paper’s scope.
>
> (4) Comments on “dimension-adaptive,” “finite-dimensional,” and “summary formulation”
>
> These terms are used in a strictly mathematical and non-asymptotic sense. By finite-dimensional and dimension adaptive we mean that all our guarantees hold for each fixed $d$ with explicit constants, rather than only asymptotically as $d \rightarrow \infty$.
>
> The term “summary formulation” does not appear anywhere in the paper.
>
> (5) Faithfulness-robustness
>
> “Faithfulness-robust” in our paper refers to the use of $\epsilon$-interventional faithfulness, which is strictly weaker than traditional (conditional-independence–based) faithfulness assumptions. In particular:
> - it does not require conditional independencies to hold exactly,
> - it tolerates violations of classical faithfulness,
> - it is preserved under latent confounding,
> - and it requires only that interventions induce detectable marginal shifts along directed paths.
>
> Thus, the guarantees remain valid even when the usual d-separation / conditional-independence structure is nearly unfaithful or completely unreliable. This robustness concerns the statistical assumptions underlying identifiability, not robustness of any optimization method.
>
>
> The technical points raised by the reviewer primarily pertain to a different class of causal-discovery papers: those proposing new permutation-based or continuous-relaxation algorithms. Our submission is a theoretical deviation analysis of an existing score under random interventions, without proposing a new algorithm or search method. We hope this clarification helps properly contextualize the contribution.

---

### Official Review · Reviewer_Do7d · 2025-10-30

**Soundness:** 4
**Presentation:** 3
**Contribution:** 3
**Rating:** 8
**Confidence:** 4

**Summary:**

This paper studies identifiability properties of causal graphs under single-node interventions from a probabilistic perspective. In particular, the authors study the number of misoriented edges (or, normalizing, the false negative rate), when the topological ordering of the graph is estimated via the InterSort algorithm.

The authors study three random graph models: dense Erdos-Renyi (ER) models, sparse ER models, and a generalized Barabasi-Albert (BA) model, and a single intervention model where each single-node intervention is present with probability $p_{\text{int}}$. For graphs on $d$ nodes, they show that the expected false negative rate (FNR) under the dense ER is $O(d^{-1})$, and for the latter two models, they show that the expected FNR is $O(1)$. Further, they provide deviation bounds on the FNR: in the dense ER model, they show that the FNR has deviations of order $O(d^{-1/2})$, in the sparse ER model, deviations are of order $O(\log(d) \cdot d^{-1/2})$, and under the BA model, the deviations are of the order $O(d^{\beta - 1/2})$, where $\beta = (1 + \kappa/m)^{-1} \in (0, 1)$ for an attractiveness parameter $\kappa > 0$ and a link-count parameter $m > 0$. These theoretical findings are corroborated by experiments on synthetic data using the DiffInterSort extension of the InterSort algorithm.

**Strengths:**

**Originality and significance:** To the best of my knowledge, this paper is the first to study deviation bounds for identifiability metrics in random causal graph models. Compared to just expectation results, these results are more informative and provide stronger guidance for downstream applications (e.g., the development of causal discovery algorithms which are targeted towards more easily identifiable cases).

**Quality and clarity:** The theoretical results are strong under the relatively weak assumptions (random intervention targets and single-node interventions only identify neighbors). The presentation is well-structured, with the results presented in a very logical order, the writing is clear, with related work being particularly well-described.

**Weaknesses:**

In my opinion, the main weakness of the paper (shared by similar papers in the area, and even related areas like average-case complexity theory) is a somewhat shaky relation between theory and practice. In the "Limitations" section, the authors acknowledge that the random graph models are somewhat realistic, and I would also prefer the paper to emphasize that the "random intervention" model may be overly pessimistic.

However, for the sake of argument, assume that both the random graph and intervention models were realistic. In practice, causal discovery is not often applied to a large number of different high-dimensional datasets with interventions, both due to a lack of interventional data and a lack of enough samples per intervention for reliability in high-dimensional settings. Hence, it is somewhat difficult to translate these forms of theoretical results into practical relevance. While the authors make a good effort to connect their theory with potential practical implications in their "Conclusion" section, I still feel that this flavor of work verges towards "theory for its own sake".

**Questions:**

**Intervention probability:** It was somewhat hard to track how the probability of having data from any particular single-node intervention plays a role in the bounds, e.g., this probability does not appear in Table 1. I see this dependence in other places (e.g. Theorem 8), but it would be nice to make this dependence more explicit and transparent throughout, as I would consider it one of the key quantities.

---

> ### Author Response · Authors · 2025-11-20
>
> We thank the reviewer for the very positive and insightful assessment and for the careful reading of the paper. We especially appreciate the recognition that deviation guarantees for identifiability metrics have been largely unexplored, and that these results can provide more informative guidance than expectation-only analyses.
>
> (1) Scope and relation between theory and practice
>
>  We agree that random-graph analyses do not model any specific real system in detail. Their purpose is instead to capture generic structural properties—such as sparsity, degree heterogeneity, and hub structure—that are shared by many large networks. In this sense, the results provide insight into which aspects of graph structure make causal orientation more or less stable, and how these effects scale with dimension.
>
> This perspective is practically relevant in two ways. First, in domains where high-dimensional interventional data is becoming increasingly common (e.g., perturbation screens or large-scale experiments), our deviation bounds describe the stability one can expect even under minimal assumptions. Second, even when real data are limited, ER and scale-free models are the standard baselines used throughout the causal-discovery literature to assess algorithmic behavior. Understanding why these families tend to yield stable or unstable orientation errors helps practitioners correctly interpret such evaluations and calibrate expectations in typical regimes.
>
> (2) Practical relevance despite limited interventional datasets
>
> We agree that large-scale interventional datasets remain limited in some application domains. However, the settings where such data is abundant—single-cell perturbation technologies, pooled CRISPR screens, high-throughput network experiments—represent exactly the high-dimensional regimes our results are designed to address. In such domains, even when the graph is not perfectly known, the ability to guarantee concentration of orientation error under minimal assumptions is crucial for assessing robustness.
>
> More generally, our results are framed as worst-case or baseline guarantees:
> whenever interventions can be optimized or adaptively chosen, one should expect smaller mean error and tighter concentration than those proved here. Thus, our analysis provides a conservative outlook around which more sophisticated strategies can be developed.
>
> (3) Clarifying the role of intervention probability $p_{int}$
>
> We appreciate the reviewer’s comment and agree that the dependence on the intervention probability could be emphasized more transparently. A subtle but important point is that most deviation results do not explicitly contain  $p_{int}$  because the arguments rely on worst-case Lipschitz bounds. In McDiarmid-type inequalities, the sensitivity constants $c_k$ capture the maximum possible change in the error when flipping any intervention indicator, and these bounds do not depend on the probability with which indicators are sampled. As a result, even though the intervention vector is random, the deviation theorems reflect a worst-case dependence structure that is independent of  $p_{int}$ .
>
> However, we agree with the reviewer that the dependence on  $p_{int}$  is important for expectation-level results and for intuition about coverage. For the bounds on the expected number of misorientations,  $p_{int}$  plays an explicit role because it controls the probability that a true parent–child relation is revealed by a random intervention. We will update these expectation results to state this dependence explicitly in Table 1.
>
> We appreciate the reviewer’s strong evaluation of the originality, soundness, and significance of the work, and we will incorporate the helpful suggestions to further clarify scope and strengthen the link to practice.

---

> > ### Comment · Reviewer_Do7d · 2025-11-23
> >
> > (1) Thanks for expanding on the practical relevance. In the revised paper, I'd be keen to see a discussion of this practical relevance in the introduction, to give readers some guidance on why to care about these kinds of results. Essentially, I see the potential "users" of these results as methodologists who wish to (a) exploit aspects of such graphical structures in algorithm design and (b) accurately assess their methods in synthetic settings.
> >
> > (2) I appreciate the framing as worst-case or baseline guarantees, perhaps the practical consequences of this framing can be slightly more emphasized (e.g. if one seeks to prove a similar result in a more optimistic setting, it should be better than this result).
> >
> > (3) Thanks for agreeing with the point about the importance of the intervention probability, and for updating the results in Table 1 to state this dependency. I appreciate the explanation about the different nature of the expectation-level vs. deviation-level results, which intuitively makes sense. This point raises an interesting question as to whether a different proof technique (something stronger than McDiarmid-type inequalities?) would give deviation-level results which also depend on the intervention probability. Perhaps this question could be included in the discussion and the lack of this dependence could be mentioned as a minor limitation.

---

> > > ### Author Response · Authors · 2025-11-25
> > >
> > > (1) Practical relevance.
> > >
> > > We thank you for highlighting the importance of clarifying the practical relevance of our results. In the revised manuscript, we added a dedicated paragraph in the introduction explaining how the deviation bounds are useful to both methodologists and practitioners.
> > >
> > > (2) Baseline versus optimistic regimes.
> > >
> > > We agree that a framing as “baseline guarantees” can be made more explicit. We have revised the introduction to highlight that our deviation results constitute the minimal level of stability one should expect under randomized single-node interventions and $\epsilon$-interventional faithfulness. As noted above, any algorithm or theory that leverages additional structure or richer interventions should achieve deviation properties that strictly improve upon these baseline bounds.
> > >
> > > (3) Dependence on the intervention probability.
> > >
> > > We now mention this in the limitations section and note that developing proof techniques capable of yielding deviation-level dependence on $p_{int}$ is an interesting direction for future work.

---

### Official Review · Reviewer_esW9 · 2025-11-04

**Soundness:** 3
**Presentation:** 3
**Contribution:** 2
**Rating:** 4
**Confidence:** 2

**Summary:**

This paper considers the problem of  causal discovery under random interventions on large random graphs. The authors study how accurately causal directions can be identified when single-variable interventions are applied to networks drawn from Erd\H{o}s--R\'{e}nyi (ER) and Barab\'{a}si--Albert (BA) models. A key contribution is the  derivation of the deviation bounds for the false-negative rate (FNR)---the fraction of causal edges whose orientation is not correctly recovered. The paper shows that, under an $\varepsilon$-interventional faithfulness assumption (milder than $d$-separation based faithfulness), the FNR not only remains small in expectation but also concentrates sharply around its mean as the number of nodes grows. For dense ER graphs, the expected FNR decays as $\Theta(1/d)$ with concentration rate $O(d^{-1/2})$; for sparse ER graphs, it stabilizes at $O(1)$ with $O((\log d)/\sqrt{d})$ deviations. The paper also give bound for and BA (scale-free) graphs, though they are bit more technical to write. The paper also empirical illustrates the predicted concentration trends.

**Strengths:**

Causal structure discovery is a fundamental and challenging problem in machine learning and scientific inference. Any progress in understanding its limits and guarantees is valuable. Thus the broad topic of the  paper is well motivated. The paper is primarily theoretical, providing detailed proofs in the appendix and clearly stated assumptions. They also give a very good description of related work. While the main results are theoretical, the authors also include empirical experiments illustrating the concentration of the FNR on both ER and BA graphs. These simulations, though modest, help validate the theoretical findings.

**Weaknesses:**

The main weakness I find is that it is not clear whether causal discovery on random graphs corresponds to realistic application domains. Most real-world causal systems are I presume are more structured rather than randomly generated. Without concrete scenarios where random-graph analysis informs practice, the practical utility of these results remains uncertain. Thus the paper would be strengthened by examples of settings where such asymptotic guarantees could guide real causal-discovery in practice. While the paper appears technically sound, I am not able to enthusiastically recommend acceptance without a clearer bridge to practical utility of the modeling. That said, I am open to other expert reviewers’ perspectives, especially those with stronger background in causal-discovery theory that the paper is addressing.

**Questions:**

My main question is related to the weakness I mentioned. Can the authors provide motivating examples where causal-discovery performance on random-graph models offers insights applicable to real-world causal networks? Are there particular domains where random interventions approximate realistic experimental designs?

---

> ### Author Response · Authors · 2025-11-20
>
> We thank the reviewer for the thoughtful and constructive feedback, and we fully agree that clarifying the connection between random-graph analysis and practical causal discovery applications strengthens the paper.
>
> (1) Why ER and BA models are relevant beyond purely theoretical interest
>
> While real-world causal networks are not literal ER or BA graphs, both families capture key structural statistics of many large causal systems. For example:
> - Generalized BA models closely approximate degree distributions of gene regulatory networks, where preferential attachment is well documented empirically through evolutionary and functional studies [1].
> - Neural connectivity networks exhibit heavy-tailed and hub-dominated structure, again aligning well with preferential-attachment behavior [2,3].
> - ER and sparse-ER models serve as canonical average-case models for high-dimensional, weakly structured systems, where only sparsity is known.
>
> Our goal is not to claim that ER/BA models are accurate mechanistic descriptions of any specific real system, but rather that they capture typical large-scale statistical properties—such as sparsity, heavy-tailed degrees, and heterogeneity—that strongly influence the difficulty of causal orientation. In this sense, they provide a principled baseline for understanding how graph structure impacts fundamental limits.
>
> (2) Random-intervention designs
>
> We emphasize that random interventions represent a worst-case or baseline scenario: whenever interventions can be deliberately designed or optimized, one should expect strictly stronger recovery guarantees than those we analyze. Our goal is therefore to provide conservative deviation bounds that hold even without any optimization of the intervention set.
>
> At the same time, random intervention policies are also common in applications, precisely because designing an effective intervention plan is often infeasible. In many high-dimensional systems—whether biological, social, or engineered—one rarely has enough prior structural knowledge to select interventions strategically. As a result, practitioners often perturb a random subset of variables, either explicitly (e.g., randomized knockouts, randomized stimulus protocols) or implicitly (e.g., passive exposure to naturally varying inputs). In such settings, the Bernoulli intervention model we analyze is a realistic representation of how interventions are actually applied.
>
>
> (3) Connection to established practice in causal-discovery evaluation
>
> We also highlight an adjacent point: ER and BA models are the standard synthetic benchmarks in causal discovery. The majority of classical and modern causal inference papers evaluate algorithms on ER, scale-free, or BA-like graphs specifically because they reflect typical structural regimes and make results comparable across methods.
> Our theoretical results therefore directly inform the expected behavior of causal discovery methods on the very simulation settings routinely used in the literature. This adds practical relevance even when the final application domain has additional structure.
>
> We hope these additions fully clarify the practical relevance of our results and strengthen the bridge between theory and real-world causal discovery.
>
> [1] Albert R. Scale-free networks in cell biology. J Cell Sci. 2005 Nov 1;118(Pt 21):4947-57. doi: 10.1242/jcs.02714. PMID: 16254242
>
> [2] Eguiluz, Victor M., et al. "Scale-free brain functional networks." Physical review letters 94.1 (2005): 018102.
>
> [3] Achard, Sophie, et al. "A resilient, low-frequency, small-world human brain functional network with highly connected association cortical hubs." Journal of Neuroscience 26.1 (2006): 63-72.

---

### Author Response · Authors · 2025-11-20
**General Comments to All Reviewers and Area Chair**

We thank the reviewers for their thoughtful and constructive feedback. We were encouraged that all reviewers found the theoretical contributions sound and the main results clear and technically correct. The discussion centered primarily on clarifying scope and strengthening motivation, rather than on issues with the results themselves. We have made the following revisions accordingly (all marked in blue in the text):

- Clarified scope and assumptions.

We added a paragraph (ll 215-225) explaining that although InterSort provides the formal vehicle, the guarantees themselves are general and apply to any method operating under the same data assumptions and intervention design. (Reviewer ifw6)

- Strengthened motivation and connection to practical settings.

We added concise discussion on why ER and generalized BA models, while stylized, capture key structural properties of many real-world networks (sparsity, heterogeneous degrees, hubs). We clarified that random interventions represent a conservative, worst-case scenario, and noted that large-scale perturbation experiments are increasingly common in e.g. biology and neuroscience. (ll 69-79, ll 487-490). (Reviewer Do7d, Reviewer esW9)

- Added additional analysis on SynTReN-style networks.

Following Reviewer ifw6’s suggestion, we performed a deviation-bound analysis on two SynTReN-based biological networks ($d=300$ and $d=1000$) using the general fixed-graph bounds. The results confirm the same dimension-adaptive concentration behavior observed in the parametric models. (Appendix H)

- Dependen on $p_{int}$

We added the dependence of the bounds on $p_{int}$ in table 1. (Reviewer Do7d)

We believe the revisions address the reviewers’ concerns.

---

### Meta-Review · Area_Chair_iFSD · 2026-01-06

**Summary:**

Reviewer esW9 is concerned with the gap between the random graph assumption vs. practical causal systems.

An expert reviewer, Reviewer Do7d, found the contribution valuable and strongly supports publication of this paper despite them also pointing out the potential gap between the random DAG model and practical causal graphs.

Reviewer A2f8 mistakenly posted their review for a different paper, so I will be ignoring their (low) score and review in my evaluation. Note that taking out their score of 2 out would significantly increase the paper's average score.

Reviewer ifw6 had a similar concern about practicality of the random causal graph assumption.

**Reviewer Concerns:**

Reviewer Do7d seems at least somewhat satisfied with the discussion. Reviewer ifw6 was still questioning why the random causal graphs studied in this paper are realistic.

**Reviewer Scores:**

I don't really think any reviewer would have changed their score based on the rebuttal. The supportive reviewer already gave an 8 that would not go up. The critical reviewers are marginally below the acceptance threshold, mainly for immutable characteristics of the paper, such as the analysis of random graphs.

---

### Decision · Program_Chairs · 2026-01-26

Accept (Poster)